# Sample Complexity of Correlation Detection in the Gaussian Wigner Model

**Dong Huang** [1]  **Pengkun Yang** [1]

## Abstract

Correlation analysis is a fundamental step in uncovering meaningful insights from complex datasets. In this paper, we study the problem of detecting correlations between two random graphs following the Gaussian Wigner model with unlabeled vertices. Specifically, the task is formulated as a hypothesis testing problem: under the null hypothesis, the two graphs are independent, while under the alternative hypothesis, they are edge-correlated through a latent vertex permutation, yet maintain the same marginal distributions as under the null. We focus on the scenario where two induced subgraphs, each with a fixed number of vertices, are sampled. We determine the optimal rate for the sample size required for correlation detection, derived through an analysis of the conditional second moment. Additionally, we propose a polynomial-time algorithm that significantly reduces running time.

## 1. Introduction

Understanding the correlation between datasets is one of the most significant tasks in statistics. In many applications, the observations may not be the familiar vectors but rather graphs. Recently, there have been many studies on the problem of detecting graph correlation and recovering the alignments of two correlated graphs. This problem arises across various domains:

- In computer vision, 3-D shapes can be represented as graphs, where nodes are subregions and weighted edges represent adjacency relationships between different regions. A fundamental task for pattern recognition and image processing is determining whether two graphs represent the same object under different rotations (Berg et al., 2005; Mateus et al., 2008).

- In natural language processing, each sentence can be represented as a graph, where nodes correspond to words or phrases, and the weighted edges represent syntactic and semantic relationships (Hughes & Ramage, 2007). The ontology alignment problem refers to uncovering the correlation between knowledge graphs that are in different languages (Haghighi et al., 2005).

- In computational biology, protein–protein interactions (PPI) and their networks are crucial for all biological processes. Proteins can be regarded as vertices, and the interactions between them can be formulated as weighted edges (Singh et al., 2008; Vogelstein et al., 2015).

Following the hypothesis testing framework proposed in Barak et al. (2019), we formulate the graph correlation detection problem in Problem 1.1. For a weighted graph $\mathbf{G}$ with vertex set $V(\mathbf{G})$ and edge set $E(\mathbf{G})$, the weight associated with each edge $uv$ is typically denoted as $\beta_{uv}(\mathbf{G})$ for any $u, v \in V(\mathbf{G})$.

*Problem* 1.1. Let $\mathbf{G}_1$ and $\mathbf{G}_2$ be two weighted random graphs with vertex sets $V(\mathbf{G}_1), V(\mathbf{G}_2)$ and edge sets $E(\mathbf{G}_1), E(\mathbf{G}_2)$. Under the null hypothesis $\mathcal{H}_0$, $\mathbf{G}_1$ and $\mathbf{G}_2$ are independent; under the alternative hypothesis $\mathcal{H}_1$, there exists a correlation between $E(\mathbf{G}_1)$ and $E(\mathbf{G}_2)$. Given $\mathbf{G}_1$ and $\mathbf{G}_2$, the goal is to test $\mathcal{H}_0$ against $\mathcal{H}_1$.

A variety of studies have extensively investigated detection problems. However, the previous studies typically required full observation of all edges in $\mathbf{G}_1$ and $\mathbf{G}_2$ for detection, which is impractical when the entire graph is unknown in certain scenarios. In such cases, graph sampling—the process of sampling a subset of vertices and edges from the graph—becomes a powerful approach for exploring graph structure. This technique has been widely used in various settings, as it allows for inference about the graph without needing full observation (Leskovec & Faloutsos, 2006; Hu & Lau, 2013). In fact, there are several motivations leading us to consider the graph sampling method:

- Lack of data. In social network analysis, the entire network is often unavailable due to API limitations.

[1]Department of Statistics and Data Science, Tsinghua University, Beijing, China. Correspondence to: Dong Huang <hd23@mails.tsinghua.edu.cn>, Pengkun Yang <yangpengkun@tsinghua.edu.cn>.

*Proceedings of the 42nd International Conference on Machine Learning*, Vancouver, Canada. PMLR 267, 2025. Copyright 2025 by the author(s).

As a result, researchers typically select a subset of users from the network, which essentially constitutes a sampling of vertices (Papagelis et al., 2011).

- Testing costs. The Protein Interaction Network is a common focus in biochemical research. However, accurately testing these interactions can be prohibitively expensive. As a result, testing methods based on sampled graphs are often employed (Stumpf et al., 2005).

- Visualization. The original graph is sometimes too large to be displayed on a screen, and sampling a subgraph provides a digestible representation, making it easier for visualization (Wu et al., 2016).

In this paper, we consider sampling induced subgraphs for testing $\mathcal{H}_0$ against $\mathcal{H}_1$ when given two random graphs $\mathbf{G}_1$ and $\mathbf{G}_2$ with $|V(\mathbf{G}_1)| = |V(\mathbf{G}_2)| = n$. We randomly sample two induced subgraphs $G_1, G_2$ with $s$ vertices from $\mathbf{G}_1$ and $\mathbf{G}_2$, respectively. An induced subgraph of a graph is formed from a subset of the vertices of the graph, along with all the edges between them from the original graph. Specifically, the sampling process for $G_1$ and $G_2$ is as follows: we first independently select vertex sets $V(G_1) \subseteq V(\mathbf{G}_1)$ and $V(G_2) \subseteq V(\mathbf{G}_2)$ with $|V(G_1)| = |V(G_2)| = s$, and then retain the weighted edge between $V(G_1)$ and $V(G_2)$ from the original graphs. We assume $s \leq n$ throughout the paper.

## 1.1. Main Results

In this subsection, we present the main results of the paper. Numerous graph models exist, with the Gaussian Wigner model being a prominent example (Ding et al., 2021; Fan et al., 2023), under which the weighted edges $\beta_{uv}(\mathbf{G})$ follow independent standard normals for any vertices $u, v \in V(\mathbf{G})$. This paper focuses on the Gaussian Wigner model with vertex set size $n$. Under the null hypothesis $\mathcal{H}_0$, $\mathbf{G}_1$ and $\mathbf{G}_2$ follow independent Gaussian Wigner model with $n$ vertices. Under the alternative hypothesis $\mathcal{H}_1$, $\mathbf{G}_1$ and $\mathbf{G}_2$ follow the following correlated Gaussian Wigner model.

**Definition 1.2** (Correlated Gaussian Wigner model). Let $\pi^*$ denote a latent bijective mapping from $V(\mathbf{G}_1)$ to $V(\mathbf{G}_2)$. We say that a pair of graphs $(\mathbf{G}_1, \mathbf{G}_2)$ are correlated Gaussian Wigner graphs if each pair of weighted edges $\beta_{uv}(\mathbf{G}_1)$ and $\beta_{\pi^*(u)\pi^*(v)}(\mathbf{G}_2)$ for any $u, v \in V(\mathbf{G}_1)$ are correlated standard normals with correlation coefficient $\rho \in (0, 1)$.

Let $\mathcal{Q}$ and $\mathcal{P}$ denote the probability measures for the sampled subgraphs $(G_1, G_2)$ under $\mathcal{H}_0$ and $\mathcal{H}_1$, respectively. We then focus on the following two detection criteria.

**Definition 1.3** (Strong and weak detection). We say a testing statistic $\mathcal{T} = \mathcal{T}(G_1, G_2)$ with a threshold $\tau$ achieves

- *strong detection*, if the sum of Type I and Type II errors

converges to 0 as $n \to \infty$:

$$\lim_{n\to\infty} [\mathcal{P}(\mathcal{T} < \tau) + \mathcal{Q}(\mathcal{T} \geq \tau)] = 0;$$

- *weak detection*, if the sum of Type I and Type II errors is bounded away from 1 as $n \to \infty$:

$$\lim_{n\to\infty} [\mathcal{P}(\mathcal{T} < \tau) + \mathcal{Q}(\mathcal{T} \geq \tau)] < 1.$$

It is well-known that the minimal value of the sum of Type I and Type II errors between $\mathcal{P}$ and $\mathcal{Q}$ is $1 - \text{TV}(\mathcal{P}, \mathcal{Q})$ (see, e.g., Polyanskiy & Wu (2025, Theorem 7.7)), achieved by the likelihood ratio test, where $\text{TV}(\mathcal{P}, \mathcal{Q}) = \frac{1}{2} \int |d\mathcal{P} - d\mathcal{Q}|$ is the total variation distance between $\mathcal{P}$ and $\mathcal{Q}$. Thus strong and weak detection are equivalent to $\lim_{n\to\infty} \text{TV}(\mathcal{P}, \mathcal{Q}) = 1$ and $\lim_{n\to\infty} \text{TV}(\mathcal{P}, \mathcal{Q}) > 0$, respectively. We then establish the main results of correlation detection in the Gaussian Wigner model.

**Theorem 1.4.** *There exist universal constants $\overline{C}, \underline{C}$ such that, for any $0 < \rho < 1$, if $s^2 \geq \overline{C} \left( \frac{n \log n}{\log(1/(1-\rho^2))} \vee n \right)$,*

$$\text{TV}(\mathcal{P}, \mathcal{Q}) \geq 0.9.$$

*Moreover, if $s^2 = \omega(n)$, $\text{TV}(\mathcal{P}, \mathcal{Q}) = 1 - o(1)$.*

*Conversely, if $s^2 \leq \underline{C} \left( \frac{n \log n}{\log(1/(1-\rho^2))} \vee n \right)$,*

$$\text{TV}(\mathcal{P}, \mathcal{Q}) \leq 0.1.$$

*Moreover, if $s^2 \leq \frac{\underline{C} n \log n}{\log(1/(1-\rho^2))}$ or $s^2 = o(n)$, $\text{TV}(\mathcal{P}, \mathcal{Q}) = o(1)$.*

The proof of Theorem 1.4 is deferred to Appendix A. Theorem 1.4 implies that, for the hypothesis testing problem between $\mathcal{H}_0$ and $\mathcal{H}_1$ when sampling two induced subgraphs, the optimal rate for the sample size $s$ is of the order $\left( \frac{n \log n}{\log(1/(1-\rho^2))} \vee n \right)^{1/2}$. Above this order, detection is possible, while below it, detection is impossible. Specifically, when $\frac{\overline{C} n \log n}{\log(1/(1-\rho^2))} > n^2$, the possibility condition requires $s > n$ in the above Theorem. However, we assume that the sample size $s \leq n$, which indicates that there is no theoretical guarantee on detection, even when we sample the entire graph. Indeed, it is shown in Wu et al. (2023) that the detection threshold on $\rho$ in the fully correlation Gaussian Wigner model is $\rho^2 \asymp \frac{\log n}{n}$. Our results match the thresholds established in the previous work up to a constant for the special case $s = n$.

The possibility results can serve as a criterion for successful correlation detection in practice. For example, in computational biology, one may sample subgraphs to reduce testing costs, and the possibility results indicate when accurate

detection remains feasible. Conversely, the impossibility results offer a theoretical tool for privacy protection. For instance, in social network de-anonymization, they imply that no test can succeed under certain conditions, thus providing a theoretical guarantee of privacy for anonymized networks.

### 1.2. Related Work

**Graph matching** The problem of graph matching refers to finding a correspondence between the nodes of different graphs (Caetano et al., 2007; Livi & Rizzi, 2013). Recently, there have been many studies on the analysis of matching two correlated random graphs. In addition to the Gaussian Wigner model, another important model is the Erdős-Rényi model (Erdős & Rényi, 1959), where the edge follows Bernoulli distribution instead of normal distribution. As shown in Cullina & Kiyavash (2016; 2017); Hall & Massoulié (2023), some sufficient and necessary conditions for the matching problem in the Erdős-Rényi model were provided. The optimal rate for graph matching in the Erdős-Rényi model has been established in Wu et al. (2022), and the constant was sharpened by analyzing the densest subgraph in Ding & Du (2023). There are also many extensions on Gaussian Wigner model and correlated Erdős-Rényi graph model, including the inhomogeneous Erdős-Rényi model (Rácz & Sridhar, 2023; Ding et al., 2023b), the partially correlated graphs model (Huang et al., 2024), the correlated stochastic block model (Chen et al., 2024; 2025), the multiple correlated graphs model (Ameen & Hajek, 2024; 2025), and the correlated random geometric graphs model (Wang et al., 2022).

**Efficient algorithms and computational hardness** There are many algorithms on the correlation detection and graph matching problem, including percolation graph matching algorithm (Yartseva & Grossglauser, 2013), subgraph matching algorithm (Barak et al., 2019), message-passing algorithm (Piccioli et al., 2022), and spectral algorithm (Fan et al., 2023), while some algorithms may be computationally inefficient. There are also many efficient algorithms, based on the different correlation coefficient, including Babai et al. (1980); Bollobás (1982); Dai et al. (2019); Ganassali & Massoulié (2020); Ding et al. (2021); Mao et al. (2023a); Ding & Li (2023); Mao et al. (2023b); Araya et al. (2024); Ding & Li (2024); Mao et al. (2024); Ganassali et al. (2024); Muratori & Semerjian (2024).

The low-degree likelihood ratio (Hopkins & Steurer, 2017; Hopkins, 2018) has emerged as a framework for studying computational hardness in high-dimensional statistical inference. It conjectures that polynomial-time algorithms succeed only in regimes where low-degree statistics succeed. Based on the low-degree conjecture, the recent work by Ding et al. (2023a); Mao et al. (2024) established sufficient conditions for computational hardness results on the

recovery and detection problems.

### 1.3. Contributions and Outlines

In this paper, we derive the optimal rate on sample size for correlation detection in the Gaussian Wigner model. Specifically, we prove that the optimal sample complexity is of rate $s \asymp \left( \frac{n \log n}{\log(1/(1-\rho^2))} \vee n \right)^{1/2}$. We also propose a polynomial algorithm that significantly reduces computational cost.

In Sections 2 and 3, we prove the possibility results and impossibility results on sample size, respectively. Section 4 introduces our polynomial algorithm for correlation detection. Then, we run some numerical experiments in Section 5 to verify the effectiveness for our algorithm proposed in Section 4. Finally, Section 6 offers further discussion and future research directions, and the appendices contain detailed proofs and additional experimental results.

## 2. Possibility Results

In this section, we prove the possibility results in Theorem 1.4 by analyzing the error probability $\mathcal{P}(\mathcal{T} < \tau) + \mathcal{Q}(\mathcal{T} \geq \tau)$ under different regimes of $\rho$, which provides an upper bound for the optimal sample complexity. Given a domain subset $S \subseteq V(G_1)$ and an injective mapping $\pi : S \mapsto V(G_2)$, along with a bivariate function $f : \mathbb{R} \times \mathbb{R} \mapsto \mathbb{R}$, we define the $f-similarity\ graph$ $\mathcal{H}_\pi^f$ as follows. The vertex set of $\mathcal{H}_\pi^f$ is $V(\mathcal{H}_\pi^f) = V(G_1)$, and for each edge $e$, the weighted edge is defined as

$$\beta_e(\mathcal{H}_\pi^f) = \begin{cases} f\left(\beta_e(G_1), \beta_{\pi(e)}(G_2)\right) & \text{if } e \in \binom{S}{2} \\ 0 & \text{otherwise} \end{cases}, \quad (1)$$

where $\pi(e)$ denotes the edge $\pi(u)\pi(v)$ for any edge $e = uv$. Let $m \triangleq \frac{(1-\epsilon)s^2}{n}$ for some constant $0 < \epsilon < 1$, and denote $\mathcal{S}_{s,m}$ as the set of injective mappings $\pi : S \subseteq V(G_1) \mapsto V(G_2)$ with $|S| = m$. Let $\mathrm{e}\left(\mathcal{H}_\pi^f\right) \triangleq \sum_{e \in E(G_1)} \beta_e\left(\mathcal{H}_\pi^f\right)$ define the sum of weighted edges in $\mathcal{H}_\pi^f$. In fact, the quantity $\mathrm{e}(\mathcal{H}_\pi^f)$ can be regarded as a similarity score between two graphs. Our test statistic takes the form

$$\mathcal{T}(f) = \max_{\pi \in \mathcal{S}_{s,m}} \mathrm{e}\left(\mathcal{H}_\pi^f\right) = \max_{\pi \in \mathcal{S}_{s,m}} \sum_{e \in E(G_1)} \beta_e\left(\mathcal{H}_\pi^f\right). \quad (2)$$

For simplicity, we write $\mathcal{T}$ for $\mathcal{T}(f)$ when the choice of $f$ is clear from the context. By the detection criteria in Definition 1.3, it suffices to bound the Type I error $\mathcal{Q}(\mathcal{T}(f) \geq \tau)$ and the Type II error $\mathcal{P}(\mathcal{T}(f) < \tau)$ for some appropriate threshold $\tau$. In the following, we outline a general recipe to derive an upper bound for error probabilities.

**Type I error.** Under the null hypothesis $\mathcal{H}_0$, the sampled subgraphs $G_1$ and $G_2$ are independent. Given a bivariate

function $f$ and a threshold $\tau$, it should be noted that the distribution of the $f-similarity$ graph $\mathcal{H}_\pi^f$ follows the same distribution for any $\pi \in \mathcal{S}_{s,m}$. Consequently, applying the union bound yields that

$$\mathcal{Q}\left(\mathcal{T} \geq \tau\right) \leq |\mathcal{S}_{s,m}| \mathcal{Q}\left(\mathsf{e}\left(\mathcal{H}_\pi^f\right) \geq \tau\right).$$

We then bound the tail probability by a standard Chernoff bound $\mathcal{Q}\left(\mathsf{e}\left(\mathcal{H}_\pi^f\right) \geq \tau\right) \leq \exp\left(-\lambda\tau\right) \mathbb{E}\left[\exp\left(\lambda \mathsf{e}\left(\mathcal{H}_\pi^f\right)\right)\right].$ See (14) in Appendix B.1 for more details.

**Type II error.** Under the alternative hypothesis $\mathcal{H}_1$, recall that $\pi^*$ denotes the latent bijective mapping from $V(\mathbf{G}_1)$ to $V(\mathbf{G}_2)$. For the induced subgraphs $G_1, G_2$ sampled from $\mathbf{G}_1, \mathbf{G}_2$, we denote the set of common vertices as

$$S_{\pi^*} \triangleq V(G_1) \cap (\pi^*)^{-1}(V(G_2)), \tag{3}$$
$$T_{\pi^*} \triangleq \pi^*(V(G_1)) \cap V(G_2). \tag{4}$$

We note that the restriction of $\pi^*$ to $S_{\pi^*}$ is a bijective mapping between $S_{\pi^*}$ and $T_{\pi^*}$, and thus $|S_{\pi^*}| = |T_{\pi^*}|$. In our random sampling models, the vertices of $G_1$ and $G_2$ are independent and identically sampled without replacement from the two graphs $\mathbf{G}_1$ and $\mathbf{G}_2$, which yields the following Lemma regarding the sizes of $S_{\pi^*}$ and $T_{\pi^*}$.

**Lemma 2.1.** *When randomly sampling vertex sets* $V(G_1), V(G_2)$ *from* $V(\mathbf{G_1}), V(\mathbf{G_2})$ *with* $|V(G_1)| = |V(G_2)| = s$, *the size of common vertex set in* (3) *follows a Hypergeometric distribution* $\mathrm{HG}(n, s, s)$. *Specifically,*

$$\mathbb{P}[|S_{\pi^*}| = t] = \binom{s}{t}\binom{n-s}{s-t} \Big/ \binom{n}{s}, \textit{ for any } t \in [s].$$

We then establish the main ingredients for controlling the Type II error. Under the distribution $\mathcal{P}$, given $f$ and $\tau$,

$$\{\mathcal{T} < \tau\} = \{\mathcal{T} < \tau, |S_{\pi^*}| < m\} \cup \{\mathcal{T} < \tau, |S_{\pi^*}| \geq m\}$$
$$\subseteq \{|S_{\pi^*}| < m\} \cup \{\mathcal{T} < \tau, |S_{\pi^*}| \geq m\}. \tag{5}$$

Since $\mathbb{E}[|S_{\pi^*}|] = \frac{s^2}{n} > m$, the first event $\{|S_{\pi^*}| < m\}$ can be bounded by the concentration inequality for Hypergeometric distribution in Lemma D.3. For the second event, it can be bounded by $\mathcal{P}\left(\mathcal{T} < \tau \,\middle|\, |S_{\pi^*}| \geq m\right)$. Under the event $\{|S_{\pi^*}| \geq m\}$, there exists $\pi_m^* \in \mathcal{S}_{s,m}$ such that $\pi_m^* = \pi^*$ on its domain set $\mathrm{dom}\,(\pi_m^*)$. The error probability of the event $\{\mathcal{T} < \tau, |S_{\pi^*}| \geq m\}$ can be bounded by $\mathcal{P}\left(\mathsf{e}\left(\mathcal{H}_{\pi_m^*}^f\right) < \tau \,\middle|\, |S_{\pi^*}| \geq m\right)$. We then use the concentration inequality to bound the tail probability. See (17) for more details.

The quantity $\mathsf{e}(\mathcal{H}_\pi^f)$ measures the similarity score of a mapping $\pi$. Under the null hypothesis, $\mathsf{e}(\mathcal{H}_\pi^f)$ has a zero mean for all $\pi$, whereas under the alternative hypothesis, its mean with $\pi = \pi_m^*$ is strictly positive owing to the underlying

correlation. We derive concentration inequalities to ensure that $\mathsf{e}\left(\mathcal{H}_{\pi_m^*}^f\right)$ exceeds the maximum spurious score arising from stochastic fluctuations under the null, as shown in Propositions 2.2 and 2.3.

## 2.1. Detection by Maximal Overlap Estimator

In this subsection, we use the test statistic (2) with $f(x, y) = xy$ for possibility results. Indeed, this estimator is equivalent to maximizing the overlap on induced subgraphs between $G_1$ and $G_2$. Specifically, we have the following Proposition.

**Proposition 2.2.** *There exists a universal constant* $C_1 > 0$ *such that, for any* $0 < \rho < 1$ *and* $\tau = \binom{m}{2}\frac{\rho}{2}$, *if* $s^2 \geq \frac{C_1 n \log n}{\rho^2}$,

$$\mathcal{P}\left(\mathcal{T} < \tau\right) + \mathcal{Q}\left(\mathcal{T} \geq \tau\right) = o(1).$$

Proposition 2.2 provides a sufficient condition on strong detection for any $0 < \rho < 1$. We refer readers to Appendix B.1 for the detailed proof. Since $1 - \mathsf{TV}(\mathcal{P}, \mathcal{Q}) \leq \mathcal{P}\left(\mathcal{T} < \tau\right) + \mathcal{Q}\left(\mathcal{T} \geq \tau\right)$, it achieves the optimal rate in Theorem 1.4 when $\rho = 1 - \Omega(1)$. However, the rate is sub-optimal when $\rho = 1 - o(1)$. In fact, $s = 2$ succeeds for detection when $\rho = 1$ by comparing the difference between all edges. We will use a new estimator in Subsection 2.2 to derive the optimal rate.

## 2.2. Detection by Minimal Mean-Squared Error Estimator

In this subsection, we use the test statistic (2) with $f(x, y) = -\frac{1}{2}(x - y)^2$ and focus on the scenario where $\rho > 1 - e^{-6}$. Indeed, this estimator is equivalent to minimizing the mean squared error between the induced subgraphs of size $m$ in $G_1$ and $G_2$, respectively. Indeed, the expected mean-square error for a correlated pair $\mathbb{E}\left[\left(\beta_e(G_1) - \beta_{\pi^*(e)}(G_2)\right)^2\right]$ is $2(1 - \rho)$, while it stays bounded away from 1 for an uncorrelated pair. As a result, the choice of $f$ effectively distinguishes between $\mathcal{H}_0$ and $\mathcal{H}_1$ under strong signal condition. We now state the following Proposition.

**Proposition 2.3.** *There exists a universal constant* $C_2 > 0$ *such that, for any* $1 - e^{-6} < \rho < 1$ *and* $\tau = 2\binom{m}{2}(\rho - 1)$, *if* $s^2 \geq C_2\left(\frac{n \log n}{\log(1/(1-\rho))} \vee n\right)$,

$$\mathcal{P}\left(\mathcal{T} < \tau\right) + \mathcal{Q}\left(\mathcal{T} \geq \tau\right) \leq 0.1.$$

*Moreover, if* $\frac{s^2}{n} = \omega(1)$, $\mathcal{P}\left(\mathcal{T} < \tau\right) + \mathcal{Q}\left(\mathcal{T} \geq \tau\right) = o(1)$.

We refer readers to Appendix B.2 for the detailed proof. Proposition 2.3 provides sufficient conditions on strong and weak detection when $\rho$ is close to 1. This result fills the gap for the optimal rate of $s$ in Proposition 2.2 when $\rho = 1 - o(1)$. In view of Propositions 2.2 and 2.3, we note

that $\rho^2 \asymp \log\left(1/(1-\rho^2)\right)$ when $0 < \rho \leq 1 - e^{-6}$ and $\log\left(1/(1-\rho)\right) \asymp \log\left(1/(1-\rho^2)\right)$ when $1 - e^{-6} < \rho < 1$. Then, there exists a universal constant $\overline{C} \geq C_1 \vee C_2$ such that

$$\frac{\overline{C}}{\log\left(1/(1-\rho^2)\right)} \geq \begin{cases} \frac{C_1}{\rho^2} & \text{if } 0 < \rho \leq 1 - e^{-6} \\ \frac{C_2}{\log(1/(1-\rho))} & \text{if } 1 - e^{-6} < \rho < 1 \end{cases}.$$

We note that $\frac{\overline{C}n\log n}{\rho^2} = \overline{C}\left(\frac{n\log n}{\rho^2} \vee n\right)$ in Proposition 2.2, and thus proving the possibility results in Theorem 1.4.

*Remark* 2.4. The possibility results can be extended to sub-Gaussian assumption on the weighted edges. The bound for the moment generating function holds under the sub-Gaussian assumption, and consequently, the Chernoff bound remains valid. See Remark B.1 for more details.

*Remark* 2.5. In the previous work (Wu et al., 2023) on the correlated Gaussian Wigner model, the correlation exists over the entire graph. The maximal overlap estimator and the minimal mean-square error estimator over two graphs are equivalent since the sum of squares of the weighted edges is fixed. However, in our sampling model, the sum of squares of the weighted edges in the two subgraphs are random variables, and thus the two estimators differ. Indeed, the Maximum Likelihood Estimator (MLE) is $\max_{\pi \in \mathcal{S}_{s,|S_{\pi^*}|}} e\left(\mathcal{H}_\pi^f\right)$ with $f(x,y) = -\rho^2(x^2 + y^2) + 2\rho xy$, where $f(x,y) \asymp \rho xy$ when $\rho = 1 - \Omega(1)$ and $f(x,y) \asymp -(x-y)^2$ when $\rho = 1 - o(1)$. The choice of different estimators reflects the use of MLE under different regimes. See (31) in Appendix C.2 for details.

## 3. Impossibility Results

In this section, we establish the impossibility results for the detection problem, which provides a lower bound on the optimal sample complexity. We first present an overview of the proof. Recall that $S_{\pi^*}$ and $T_{\pi^*}$ are the sets of common vertices defined in (3) and (4), respectively. Under our sampling model, there exists a latent mapping between $S_{\pi^*}$ and $T_{\pi^*}$ under the hypothesis $\mathcal{H}_1$. When equipped with the additional knowledge of the common vertex sets, our problem can be reduced to detection with full observations on smaller correlated Gaussian Wigner model, the detection threshold for which is established in Wu et al. (2023). As shown in Lemma 2.1, the size of $S_{\pi^*}$ and $T_{\pi^*}$ follows a hypergeometric distribution. Using the concentration inequality (36), the size of $S_{\pi^*}$ satisfies $|S_{\pi^*}| \leq (1+\epsilon)\mathbb{E}\left[|S_{\pi^*}|\right]$ with high probability. Therefore, the impossibility results from the previous work on full observations remain valid when the number of correlated nodes is substituted with $(1+\epsilon)\mathbb{E}\left[|S_{\pi^*}|\right]$. However, such a reduction only proves tight when the correlation is weak. We will establish the remaining regimes by the conditional second moment method.

For notational simplicity, we use $\mathsf{TV}(\mathcal{P}, \mathcal{Q})$ to denote $\mathsf{TV}(\mathcal{P}(G_1, G_2), \mathcal{Q}(G_1, G_2))$ in this paper. By Tsybakov (2009, Equation 2.27), the total variation distance between $\mathcal{P}$ and $\mathcal{Q}$ can be upper bounded by the second moment:

$$\mathsf{TV}\left(\mathcal{P}, \mathcal{Q}\right) \leq \sqrt{\mathbb{E}_\mathcal{Q}\left(\frac{\mathcal{P}}{\mathcal{Q}}\right)^2 - 1}. \tag{6}$$

The likelihood ratio is defined as

$$\frac{\mathcal{P}(G_1, G_2)}{\mathcal{Q}(G_1, G_2)} = \frac{1}{n!} \sum_{\pi \in \mathcal{S}_n} \frac{\mathcal{P}(G_1, G_2|\pi)}{\mathcal{Q}(G_1, G_2)}, \tag{7}$$

where $\mathcal{S}_n$ denotes the set of mappings $\pi : V(\mathbf{G}_1) \mapsto V(\mathbf{G}_2)$ between two original graphs. Note that sometimes certain rare events under $\mathcal{P}$ can cause the unconditional second moment to explode, while $\mathsf{TV}(\mathcal{P}, \mathcal{Q})$ remains bounded away from one. To circumvent such catastrophic events, we can compute the second moment conditional on such events.

We consider the following event:

$$\mathcal{E} \triangleq \left\{ (G_1, G_2, \pi) : |\pi(V(G_1)) \cap V(G_2)| \leq \frac{(1+\epsilon)s^2}{n} \right\}. \tag{8}$$

By Lemma 2.1, the size of common vertex set $|\pi(V(G_1)) \cap V(G_2)|$ follows hypergeometric distribution $\mathsf{HG}(n, s, s)$ under $\mathcal{P}$. In this paper, we define the conditional distribution as $\mathcal{P}'(G_1, G_2, \pi) = \mathcal{P}(G_1, G_2, \pi|\mathcal{E})$. By Lemma D.3, we have $\mathcal{P}(\mathcal{E}) = o(1)$ when $s = \omega\left(n^{1/2}\right)$. Using $\mathsf{TV}(\mathcal{P}, \mathcal{Q}) \leq \mathsf{TV}(\mathcal{P}', \mathcal{Q}) + o(1)$ and applying (6) on $\mathcal{P}'$ and $\mathcal{Q}$ yields that a sufficient condition for $\mathsf{TV}(\mathcal{P}, \mathcal{Q}) = o(1)$ is $\mathbb{E}_\mathcal{Q}\left(\frac{\mathcal{P}'}{\mathcal{Q}}\right)^2 = 1 + o(1)$. See (25) for more details.

### 3.1. Weak correlation

In this subsection, we present the impossibility results for weak correlation regime where $0 < \rho^2 < n^{-1/2}$.

**Proposition 3.1.** *For any* $0 < \rho^2 < n^{-1/2}$, *if* $s^2 \leq \frac{n\log n}{2\log(1/(1-\rho^2))}$, *then* $\mathsf{TV}(\mathcal{P}, \mathcal{Q}) = o(1)$.

We note that the total variation distance monotonically increases by the sample size $s$. In view of Proposition 3.1, we only need to tackle with the situation $s^2 = \frac{n\log n}{2\log(1/(1-\rho^2))}$, where $s = \omega\left(n^{1/2}\right)$ since $\rho^2 < n^{-1/2}$. Therefore, a sufficient condition for $\mathsf{TV}(\mathcal{P}, \mathcal{Q}) = o(1)$ is $\mathsf{TV}(\mathcal{P}', \mathcal{Q}) = o(1)$ by the triangle inequality. The proof of $\mathsf{TV}(\mathcal{P}', \mathcal{Q}) = o(1)$ can be reduced to the lower bound in Wu et al. (2023) using a data processing inequality when given the common vertex sets $S_{\pi^*}$ and $T_{\pi^*}$. Under weak correlation, the bottleneck is detecting the existence of latent mapping $\pi^*$. The detection is impossible even with the additional knowledge on the location of common vertices. The detailed proof of Proposition 3.1 is deferred to Appendix B.3.

## 3.2. Strong correlation

In this subsection, we present the impossibility results for strong correlation graphs where $n^{-1/2} \leq \rho^2 < 1$. Let $\tilde{\pi}$ be an independent copy of $\pi$. A key ingredient in the analysis of conditional second moment is the analysis of $\frac{\mathcal{P}(G_1,G_2|\pi)}{\mathcal{Q}(G_1,G_2)} \frac{\mathcal{P}(G_1,G_2|\tilde{\pi})}{\mathcal{Q}(G_1,G_2)}$. We refer readers to Appendix B.4 for the details.

We then analyze the terms $\frac{\mathcal{P}(G_1,G_2|\pi)}{\mathcal{Q}(G_1,G_2)}$ and $\frac{\mathcal{P}(G_1,G_2|\tilde{\pi})}{\mathcal{Q}(G_1,G_2)}$. Recall the common vertex sets $S_\pi$ and $T_\pi$ defined in (3) and (4), respectively. For any $e \notin \binom{S_\pi}{2}$ and $e' \notin \binom{T_\pi}{2}$, $\beta_e(G_1)$ and $\beta_{e'}(G_2)$ are independent under $\mathcal{P}$, while under the null hypothesis distribution $\mathcal{Q}$ they are also independent. Therefore, the term $\frac{\mathcal{P}(G_1,G_2|\pi)}{\mathcal{Q}(G_1,G_2)}$ can be decomposed into $\prod_{e \in \binom{S_\pi}{2}} \ell(\beta_e(G_1), \beta_{\pi(e)}(G_2))$, where $\ell(a,b) \triangleq \frac{\mathcal{P}(\beta_e(G_1)=a,\beta_{\pi(e)}(G_2)=b)}{\mathcal{Q}(\beta_e(G_1)=a,\beta_{\pi(e)}(G_2)=b)}$ for any $a, b \in \mathbb{R}$ is the ratio of density functions. We note that there are correlations between $\binom{S_\pi}{2}, \binom{S_{\tilde{\pi}}}{2}, \binom{T_\pi}{2}$ and $\binom{T_{\tilde{\pi}}}{2}$, yielding that $\frac{\mathcal{P}(G_1,G_2|\pi)}{\mathcal{Q}(G_1,G_2)}$ and $\frac{\mathcal{P}(G_1,G_2|\tilde{\pi})}{\mathcal{Q}(G_1,G_2)}$ are correlated. To deal with the correlation, our main idea is to decompose the edge sets into independent parts. To formally describe all correlation relationships, we use the *correlated functional digraph* of two mappings $\pi$ and $\tilde{\pi}$ between a pair of graphs introduced in Huang et al. (2024).

**Definition 3.2** (Correlated functional digraph). Let $\pi$ and $\tilde{\pi}$ be two bijective mappings between $V(\mathbf{G}_1)$ and $V(\mathbf{G}_2)$ and $S_\pi, T_\pi, S_{\tilde{\pi}}, T_{\tilde{\pi}}$ be the sets of common vertex defined in (3) and (4). The correlated functional digraph of the functions $\pi$ and $\tilde{\pi}$ is constructed as follows. Let the vertex sets be $\binom{S_\pi}{2} \cup \binom{S_{\tilde{\pi}}}{2} \cup \binom{T_\pi}{2} \cup \binom{T_{\tilde{\pi}}}{2}$. We first add every edge $e \mapsto \pi(e)$ for $e \in \binom{S_\pi}{2}$, and then merge each pair of nodes $(e, \tilde{\pi}(e))$ for $e \in \binom{S_{\tilde{\pi}}}{2}$ into one node.

After merging all pairs of nodes, the degree of each vertex in the correlated functional digraph is at most two. Therefore, the connected components of the correlated functional digraph consist of paths and cycles. For example, for a path $(e_1, \pi(e_1), \cdots, e_j, \pi(e_j))$, where $e_1, \cdots, e_j$ are edges in $G_1$, we have $\tilde{\pi}(e_2) = \pi(e_1), \cdots, \tilde{\pi}(e_j) = \pi(e_{j-1})$; for a cycle $(e_1, \pi(e_1), \cdots, e_j, \pi(e_j))$, we have $\tilde{\pi}(e_2) = \pi(e_1), \cdots, \tilde{\pi}(e_j) = \pi(e_{j-1}), \tilde{\pi}(e_1) = \pi(e_j)$. By decomposing the connected components, the analysis of edge sets is separated into independent parts. Let P and C denote the collections of vertex sets belonging to different connected paths and cycles, respectively. For any $P \in$ P and $C \in$ C, we define $\ell_e^\pi(G_1, G_2) = \ell(\beta_e(G_1), \beta_{\pi(e)}(G_2))$ and

$$L_P \triangleq \prod_{e \in \binom{S_\pi}{2} \cap P} \ell_e^\pi(G_1, G_2) \prod_{e \in \binom{S_{\tilde{\pi}}}{2} \cap P} \ell_e^{\tilde{\pi}}(G_1, G_2),$$

$$L_C \triangleq \prod_{e \in \binom{S_\pi}{2} \cap C} \ell_e^\pi(G_1, G_2) \prod_{e \in \binom{S_{\tilde{\pi}}}{2} \cap C} \ell_e^{\tilde{\pi}}(G_1, G_2).$$

Note that the sets from P and C are disjoint. Consequently, for any $P, P' \in$ P and $C, C' \in$ C, $L_P, L_{P'}, L_C$ and $L_{C'}$ are mutually independent. Furthermore, the expectations of $L_P$ and $L_C$ can be derived from the following Lemma.

**Lemma 3.3.** *For any* $P \in$ P$, C \in$ C*, we have* $\mathbb{E}_\mathcal{Q}(L_P) = 1$ *and* $\mathbb{E}_\mathcal{Q}(L_C) = \frac{1}{1-\rho^{2|C|}}$.

By Lemma 3.3 and the joint independence between different paths and cycles, we have

$$\mathbb{E}_\mathcal{Q} \left[ \frac{\mathcal{P}(G_1, G_2|\pi)}{\mathcal{Q}(G_1, G_2)} \frac{\mathcal{P}(G_1, G_2|\tilde{\pi})}{\mathcal{Q}(G_1, G_2)} \right]$$
$$= \mathbb{E}_\mathcal{Q} \left[ \prod_{P \in \mathsf{P}} L_P \prod_{C \in \mathsf{C}} L_C \right] = \prod_{C \in \mathsf{C}} \left( \frac{1}{1-\rho^{2|C|}} \right). \quad (9)$$

The cycles set C plays a key role in the analysis of conditional second moment. In order to analyze the properties of C in depth, for any $\pi$ and $\tilde{\pi}$, we define the *core set* as

$$I^* \triangleq I^*(\pi, \tilde{\pi}) \triangleq \cup_{C \in \mathsf{C}} \cup_{e \in C} \cup_{v \in V(e) \cap V(G_1)} v, \quad (10)$$

where $V(e)$ denotes the two vertices of edge $e$. Indeed, $I^*$ is the intersection set between $V(G_1)$ and all the vertices of edges in cycle set C. In fact, the quantity $\prod_{C \in \mathsf{C}} \left( \frac{1}{1-\rho^{2|C|}} \right)$ relies significantly on $I^*$. We then show the following lemma on the properties of $I^*$.

**Lemma 3.4** (Properties of the *core set*). *For* $I^*$ *in* (10) *and any* $t \leq s$*, we have*

$$I^* = \operatorname*{argmax}_{I \subseteq V(G_1), \pi(I) = \tilde{\pi}(I)} |I|, \quad \mathbb{P}[|I^*| = t] \leq \left( \frac{s}{n} \right)^{2t}.$$

We then propose the following Proposition.

**Proposition 3.5.** *For any* $n^{-1/2} \leq \rho^2 < 1$*, if* $s^2 \leq \frac{n \log n}{8 \log(1/(1-\rho^2))}$*, then* $\mathsf{TV}(\mathcal{P}, \mathcal{Q}) = o(1)$.

The detailed proof of Proposition 3.5 is deferred to Appendix B.4. In the proof, we apply the conditional second moment method with the conditional distribution $\mathcal{P}' = \mathcal{P}(\cdot|\mathcal{E})$, where $\mathcal{E}$ is defined in (8). The analysis of the conditional second moment relies significantly on the decomposition of cycles and paths of a correlated functional digraph. By Lemma 3.3, the conditional second moment can be reduced to the calculation on cycles, while the vertex set induced by all cycles is exactly $I^*$. Combining this with the properties of $I^*$ in Lemma 3.4, we finish the proof of Proposition 3.5. In fact, under the strong correlation condition, detecting $\pi^*$ is no longer the bottleneck. We instead use a more delicate analysis based on the conditional second moment method.

By (36) in Lemma D.3, there exists $\underline{C} \leq \frac{1}{8}$ such that, when $s^2 \leq \underline{C}n$, we have $\mathbb{P}[|S_{\pi^*}| = 0] \geq 0.9$, which implies

that $\mathsf{TV}(\mathcal{P}, \mathcal{Q}) \leq 0.1$. Specifically, when $s^2 = o(n)$, $\mathbb{P}[|S_{\pi^*}| = 0] = 1 - o(1)$, and thus $\mathsf{TV}(\mathcal{P}, \mathcal{Q}) = o(1)$. Combining this with Propositions 3.1 and 3.5, we prove the impossibility results in Theorem 1.4.

*Remark* 3.6. The second moment under our induced subgraph sampling model is equivalent to that on the vertex set induced by $I^*$. When fixing $I^*$, it is equal to the second moment of correlated Gaussian Wigner model with $\pi : I^* \to \pi(I^*)$. However, $I^* = I^*(\pi, \tilde{\pi})$ is a random variable of $\pi, \tilde{\pi}$, and hence a more thorough analysis on $I^*$ is needed, as shown in Lemma 3.4.

# 4. Algorithm

In this section, we present an efficient algorithm for detection. In Theorem 1.4, we show that the estimator (2) achieves the optimal sample complexity for correlation detection. However, the estimator requires searching over $\mathcal{S}_{s,m}$, with time complexity $\binom{s}{m}^2 \cdot m!$, resulting in poor performance for large graphs. Next, we propose an efficient algorithm to approximate the estimator in (2).

When the full observations of the graphs are given, there are many different efficient algorithms for detecting correlation and recovering graph matching. For instance, it is shown in Mao et al. (2023b; 2024) that counting trees is an efficient way to detect correlation and recover graph matching when the correlation coefficient $\rho > \sqrt{\alpha}$, where $\alpha \approx 0.338$ is Otter's constant introduced in Otter (1948). The message-passing algorithm (Piccioli et al., 2022; Ganassali et al., 2024) is also an efficient algorithm in the Erdős-Rényi model, which makes substantial use of the local tree structure. Another approach for graph matching is relaxing the original problem to a convex optimization problem (Fan et al., 2023). Additionally, there are approaches based on initial seeds (Mossel & Xu, 2020) and iterative methods (Ding & Li, 2024) addressing this problem.

However, for the partial alignment problem and partial correlation detection problem, where only part of the original graphs are given, it becomes more challenging to find an efficient algorithm. One approach is to use deep learning techniques (Jiang et al., 2022; Wang et al., 2023; Ratnayaka et al., 2024), while another way is to use low-degree structures, such as cliques or trees (Sharma et al., 2018). In this paper, we propose an algorithm that finds the initial seeds by matching the cliques, and then iteratively constructs the remaining mapping. Three main components of our algorithm are outlined as follows.

- *Match the small cliques.* Given two graphs $G_1$, $G_2$ and integers $K_1, N_1, N_2$ and a bivariate function $f$, we first randomly pick $N_1$ vertex set $V_1, \cdots, V_{N_1} \subseteq V(G_1)$ with $|V_1| = \cdots = |V_{N_1}| = K_1$ and $V_i \neq V_j$, for any $1 \leq i < j \leq N_1$. For any $1 \leq i \leq N_1$, define the

injection $\pi_i : V_i \mapsto V(G_2)$ as

$$\pi_i \triangleq \operatorname*{argmax}_{\substack{\pi : V_i \mapsto V(G_2) \\ \pi \text{ injection}}} \sum_{e \in \binom{V_i}{2}} \beta_e \left( \mathcal{H}^f_\pi \right), \qquad (11)$$

where $\mathcal{H}^f_{\pi_i}$ is the $f-$*similarity graph* defined in (1). We then sort the values $\sum_{e \in \binom{V_i}{2}} \beta_e \left( \mathcal{H}^f_{\pi_i} \right)$ in decreasing order and select the top $N_2$ corresponding pairs of $(V_i, \pi_i)$. Without loss of generality, we assume that $(V_1, \pi_1), \cdots, (V_{N_2}, \pi_{N_2})$ are the top $N_2$ pairs.

- *Find seeds.* Given an integer $K_2$, for any $U \subseteq [N_2]$ with $|U| = K_2$, let $V_U \triangleq \cup_{j \in U} V_j$. We say $U$ is *compatible* if for any $v \in V_U$, $\pi_j(v)$'s are identical for all $j \in U$ such that $v \in V_{i_j}$. Let $\mathcal{I}(U)$ denote the indicator function of *compatible* set $U$. If $\mathcal{I}(U) = 1$, we define $\pi_U$ as the union of $\pi_j$ for any $j \in U$. Specifically,

$$\pi_U(v) = \pi_j(v) \text{ such that } v \in V_{i_j}, \text{ for any } v \in V_U.$$

The seed is then defined as

$$\pi_0 = \operatorname*{argmax}_{\pi_U : \mathcal{I}(U) = 1, U \subseteq [N_2], |U| = K_2} \sum_{e \in \binom{V_U}{2}} \frac{\beta_e \left( \mathcal{H}^f_{\pi_U} \right)}{\binom{|V_U|}{2}}, \qquad (12)$$

which maximize the average similarity score over $U$.

- *Iteratively construct mappings.* Define the domain set and image set of $\pi_0$ as $S_0$ and $T_0$, respectively. Then, we have $\pi_0 : S_0 \subseteq V(G_1) \mapsto T_0 \subseteq V(G_2)$. Next, we iteratively extend the seed mapping by finding one vertex each from $V(G_1)$ and $V(G_2)$ until $|S_0| = |T_0| = m$. Specifically, given $\pi_0 : S_0 \mapsto T_0$, let

$$v_1, v_2 = \operatorname*{argmax}_{\substack{v_1 \in V(G_1) \setminus S_0 \\ v_2 \in V(G_2) \setminus T_0}} \sum_{v \in S_0} f \left( \beta_{v_1 v}(G_1), \beta_{v_2 \pi_0(v)}(G_2) \right).$$

Then, we add the new mapping $v_1 \mapsto v_2$ to $\pi_0$. This process is repeated iteratively, updating $\pi_0$ until $|S_0| = m$. Finally, we compute the test statistic $\sum_{e \in \binom{s_0}{2}} \beta_e \left( \mathcal{H}^f_{\pi_0} \right)$. $\mathcal{H}_0$ is rejected if the test statistic exceeds the given threshold $\tau$, otherwise $\mathcal{H}_0$ is accepted.

The detailed algorithm is shown in Algorithm 1. Our algorithm comprises three main steps. In the first step, we select $N_1$ vertex sets $V_1, \cdots, V_{N_1}$ of size $K_1$ and search for injections $\pi_i$ from $V_i$ to $V(G_2)$, which requires $O(N_1 \cdot s^{K_1})$ time. In the second step, we search over all subsets $U \subseteq [N_2]$ with $|U| = K_2$, which takes $O(N_2^{K_2})$ time. In the third step, we iteratively expand the mapping based on our seeds, which takes $O(m^2 s^2)$ time. We typically choose $N_1 \asymp s^{K_1}$

and $K_1 \geq 3$, and thus the overall time complexity of the algorithm is $O(N_1 \cdot s^{K_1} + N_2^{K_2})$.

Since only partial correspondence exists between the two graphs under $\mathcal{H}_1$, finding the true mapping is challenging. We first use small cliques of size $K_1$ to trade accuracy for computational efficiency, although this often results in many incorrect mappings. To improve accuracy, we then test the compatibility of these small mappings and merge $K_2$ of them to construct a larger, more accurate mapping. This larger mapping is then used as a seed, and we iteratively enlarge it by adding one pair at a time until the size reaches $m$. This approach significantly reduces the running time compared to directly matching the larger cliques.

As for the performance, a larger sample size $s$ leads to larger common vertex sets, and thus increases the number of correct mappings in Step 1. A larger $K_1$ corresponds to matching larger cliques in the first step. This increases the proportion of correct mappings within the $N_2$ candidate pairs when $K_1$ is below the size of common vertex sets. However, choosing $K_1$ beyond this size introduces wrong mappings. Besides, in the second step, we search over all $U \subseteq [N_2]$ with $|U| = K_2$ to identify the seeds. While a larger $K_2$ imposes a stricter matching criterion, choosing $K_2$ beyond the number of available correct mappings from Step 1 will degrade performance.

The accuracy and running time depend on $N_1, N_2, K_1$, and $K_2$, and there is a trade-off between them: larger values of these parameters generally improve accuracy but increase the computational cost.

## 5. Numerical Experiments

In this section, we provide numerical results for Algorithm 1 on synthetic data. To this end, we independently generate 100 pairs of graphs that follow the independent Gaussian Wigner model, and another 100 pairs that follow the correlated Gaussian Wigner model with correlation $\rho$.

Fix $n = 50, s = 25, \rho = 0.99, K_1 = 4, K_2 = 3, N_1 = 10000, N_2 = 500$, and $\epsilon = 0.01$. Then, $m = \left\lfloor \frac{(1-\epsilon)s^2}{n} \right\rfloor = 12$. In Figure 1, we plot the histogram of our approximated estimator $\sum_{e \in \binom{s_0}{2}} \beta_e \left(\mathcal{H}_{\pi_0}^f\right)$ defined in Algorithm 1. We see that the histograms under the independent model and the correlated model are well-separated. By picking an appropriate threshold $\tau$, the proposed algorithm succeeds in correlation detection. We note that when $K_1 = 2$ and $K_2 = 1$, Algorithm 1 is equivalent to comparing the pairwise differences of all edges, while our approach with $K_1 = 4$ and $K_2 = 3$ is more effective than this trivial method.

In order to compare our test statistic under different settings, we plot the Receiver Operating Characteristic (ROC) curves by varying the detection threshold and plotting the

---

**Algorithm 1** Clique-Based Detection Algorithm

1: **Input:** Two graphs $G_1, G_2$ with $s$ vertices, mapping size $m$, clique size $K_1$, combining size $K_2$, number of cliques $N_1$, number $N_2$, threshold $\tau$.
2: **Output:** Detection result $\mathcal{H}_0$ or $\mathcal{H}_1$.
3: Randomly select $N_1$ vertices sets $V_i \subseteq V(G_1)$ with $|V_i| = K_1$, for any $i = 1, 2, \cdots, N_1$.
4: For each $V_i$, compute $\pi_i$ according to (11). Then, sort the values $\sum_{e \in \binom{V_i}{2}} \beta_e \left(\mathcal{H}_{\pi_i}^f\right)$ in descending order and select the top $N_2$ corresponding pairs of $(V_i, \pi_i)$. Without loss of generality, denote pairs as $(V_1, \pi_1), \cdots, (V_{N_2}, \pi_{N_2})$.
5: Find the seed mapping $\pi_0 : S_0 \subseteq V(G_1) \mapsto T_0 \subseteq V(G_2)$ according to (12).
6: **while** $|S_0| < m$ **do**
7:     **for** $v_1 \in V(G_1) \backslash S_0$ and $v_2 \in V(G_2) \backslash T_0$ **do**
8:         Compute $\sum_{v \in S_0} f \left(\beta_{v_1 v}(G_1), \beta_{v_2 \pi_0(v)}(G_2)\right)$.
9:     **end for**
10:     Find the pair $(v_1, v_2)$ for the maximal value of $\sum_{v \in S_0} f \left(\beta_{v_1 v}(G_1), \beta_{v_2 \pi_0(v)}(G_2)\right)$ and add $v_1 \mapsto v_2$ into $\pi_0$.
11: **end while**
12: Compute $\sum_{e \in \binom{s_0}{2}} \beta_e \left(\mathcal{H}_{\pi_0}^f\right)$, output $\mathcal{H}_1$ if it exceeds $\tau$, otherwise output $\mathcal{H}_0$.

---

Type II error against the Type I error. We also compute the area under the curve (AUC), which can be interpreted as the probability that the test statistic is larger for a pair of correlated graphs than a pair of independent graphs.

In Figure 2, for each plot, we fix $n = 50, \rho = 0.98, K_1 = 4, K_2 = 3, N_1 = 10000, N_2 = 500, \epsilon = 0.01$, and vary $s \in \{10, 20, 30, 40, 50\}$, with $m = \left\lfloor \frac{(1-\epsilon)s^2}{n} \right\rfloor$. We observe that as $s$ increases, the ROC curve is moving toward the upper left corner, and the AUC increases from 0.52 to 1, indicating an improvement in the performance of our test statistic. Indeed, by Lemma 2.1, the cardinality of common set increases as $s$ increase, strengthening the signal and facilitating correlation detection.

In Figure 3, for each plot, we fix $n = 50, s = 40, K_1 = 4, K_2 = 3, N_1 = 10000, N_2 = 500, \epsilon = 0.01$, and vary $\rho \in \{0.95, 0.96, 0.97, 0.98, 0.99\}$, with $m = \left\lfloor \frac{(1-\epsilon)s^2}{n} \right\rfloor = 31$. We observe that as $\rho$ increases, the ROC curve is moving toward the upper left corner, and the AUC increases from 0.55 to 1, indicating an improvement in the performance of our test statistic as the correlation strengthens. It turns out that correlation detection improves as $s$ and $\rho$ increase.

We also compare our method with the classical Graph Edit Distance (GED) (Sanfeliu & Fu, 1983), a widely used graph similarity measure. When $n = 50, s = 30$, and $\epsilon = 0.01$, the AUC values for the GED-based test at $\rho = 0.98, 1 -$

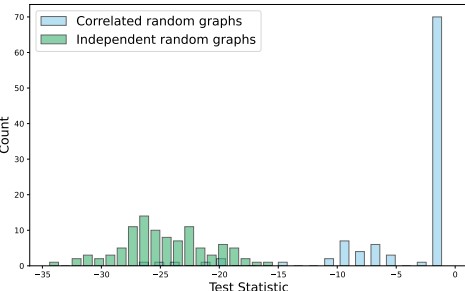

*Figure 1.* The histogram of the approximate test statistic $\sum_{e \in \binom{s_0}{2}} \beta_e \left( \mathcal{H}_{\pi_0}^f \right)$ in Algorithm 1 over 100 pairs of graphs, where the blue one represents the correlated Gaussian Wigner model, and the green one represents the independent graphs.

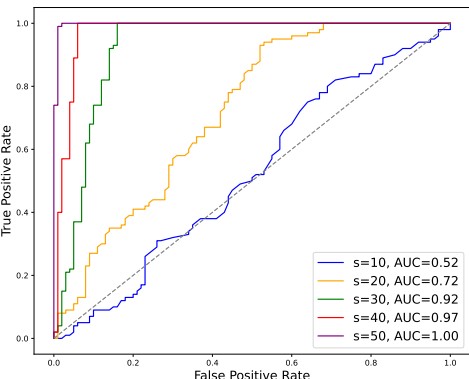

*Figure 2.* Comparison for the ROC curve of the approximate test statistic for different sample size $s$.

$10^{-6}, 1 - 10^{-7}$ are $0.53, 0.73$, and $0.88$, respectively. In contrast, our algorithm yields significantly higher AUCs of $\rho$ are $0.92, 1, 1$ under the same settings. These results demonstrate the superior performance of our method in detecting correlation in the Gaussian Wigner model. We provide some additional experiments in Appendix E.

## 6. Future Directions and Discussions

This paper focuses on detecting correlation in the Gaussian Wigner model by sampling two induced subgraphs from the original graphs. We determine the optimal rate on the sample size for correlation detection. In comparison to detection problem on the fully correlated Gaussian Wigner model, the additional challenge arises from partial correlation when sampling subgraphs. We provide a detailed analysis of the *core set* when using the conditional second moment method to derive the impossibility results. We find that the conditional second moment can be reduced to the second moment on the *core set*. Additionally, we propose an efficient approximate algorithm for correlation detection based on the clique mapping technique and an iterative approach. There

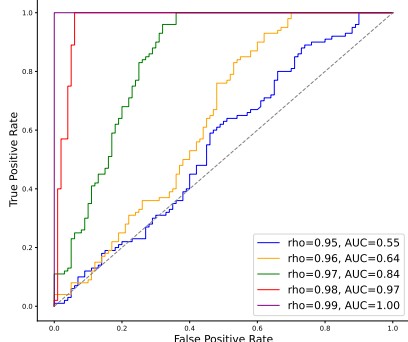

*Figure 3.* Comparison for the ROC curve of the approximate test statistic for different correlation coefficients $\rho$.

are many problems to be further investigated:

- *Extension to Erdős-Rényi model.* Most results in this paper can be extended to the Erdős-Rényi model. The key difference lies in the additional parameter $p$ controlling the edge connection probability. For the possibility results, the estimator is similar to (2), with the bivariate function $f$ selected via MLE under the Erdős-Rényi model. For the impossibility results, the reduction procedure provides tight bounds when $p = n^{-\Omega(1)}$, and a more delicate event is required for the conditional second moment analysis when $p = n^{-o(1)}$, which is similar to Proposition 3.5.

- *Theoretical analysis of the efficient algorithm.* We have shown that the Algorithm 1 performs well on synthetic data, while the theoretical guarantee remains an open problem. This guarantee can serve as an upper bound for the existence of a polynomial-time algorithm. Moreover, since the tree-counting-based method shows strong performance in the Erdős-Rényi model, it would be interesting to investigate whether it remains effective in Gaussian networks.

- *Computational hardness.* The low-degree conjecture has recently provided evidence of the computational hardness on related problems (see, e.g., Hopkins (2018); Kunisky et al. (2019)). It is of interest to investigate the computational hardness conditions with respect to the sample size for the correlation detection problem using the low-degree conjecture.

- *Other graph models.* The sample complexity for correlation detection remains unknown for many models (e.g., the stochastic block model, the graphon model). A natural next step is to explore whether our results can be extended to more general settings.

## Acknowledgments

The authors thank all reviewers for their valuable comments and suggestions. P. Yang is supported in part by the National Key R&D Program of China 2024YFA1015800.

## Impact Statement

This paper addresses the sample complexity of the correlation detection problem in the Gaussian Wigner model, with implications for fields like social network analysis, biology, and natural language processing. By improving the ability to detect correlations in complex datasets, our work contributes to advancing the understanding of hidden structures in large-scale data. The proposed methods have the potential to enhance analytical capabilities across a range of applications, enabling more accurate inferences in real-world contexts. To the best of our knowledge, there is no negative societal impact related to this research.

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

## A. Proof of Theorem 1.4

For the possibility results, by Propositions 2.2 and 2.3, if

$$s^2 \geq \begin{cases} \frac{C_1 n \log n}{\rho^2} & 0 < \rho \leq 1 - e^{-6} \\ C_2 \left( \frac{n \log n}{\log(1/(1-\rho))} \vee n \right) & 1 - e^{-6} < \rho < 1 \end{cases},$$

then $\mathsf{TV}\,(\mathcal{P}, \mathcal{Q}) \leq 0.1$. Furthermore, if $s^2 = \omega(n)$, then $\mathsf{TV}(\mathcal{P}, \mathcal{Q}) = o(1)$. Since $\frac{\log(1/(1-\rho^2))}{\rho^2} \leq \frac{\rho^2/(1-\rho^2)}{\rho^2} = \frac{1}{1-\rho^2} \leq \frac{1}{1-(1-e^{-6})^2}$ for any $0 < \rho \leq 1 - e^{-6}$, we obtain that $\mathsf{TV}\,(\mathcal{P}, \mathcal{Q}) \leq 0.1$ when $s^2 \geq \frac{C_1}{1-(1-e^{-6})^2} \cdot \frac{n \log n}{\log(1/(1-\rho^2))}$. Since $\frac{\log(1/(1-\rho^2))}{\log(1/(1-\rho))} = 1 + \frac{\log(1/(1+\rho))}{\log(1/(1-\rho))} \leq 2$ for any $1 - e^{-6} < \rho < 1$, it follows that when $s^2 \geq 2C_2 \left( \frac{n \log n}{\log(1/(1-\rho))} \vee n \right)$, $\mathsf{TV}\,(\mathcal{P}, \mathcal{Q}) \leq 0.1$. Let $\overline{C} = \frac{C_1}{1-(1-e^{-6})^2} \vee 2C_2$. Then, for $s^2 \geq \overline{C} \left( \frac{n \log n}{\log(1/(1-\rho^2))} \vee n \right)$, we have $\mathsf{TV}\,(\mathcal{P}, \mathcal{Q}) \leq 0.1$.

For the impossibility results, by Propositions 3.1 and 3.5, if $s^2 \leq \frac{n \log n}{8 \log(1/(1-\rho^2))}$, then $\mathsf{TV}\,(\mathcal{P}, \mathcal{Q}) = o(1)$. According to the concentration inequality (36) for the Hypergeometric distribution in Lemma D.3, there exists a constant $\underline{C} \leq \frac{1}{8}$ such that, when $s^2 \leq \underline{C}n$, we have $\mathbb{P}\,[|S_{\pi^*}| \geq 1] \leq 0.1$, implying $\mathbb{P}\,[|S_{\pi^*}| = 0] \geq 0.9$ and thus $\mathsf{TV}\,(\mathcal{P}, \mathcal{Q}) \leq 0.1$. Additionally, when $s^2 = o(n)$, we have $\mathbb{P}\,[|S_{\pi^*}| = 0] = 1 - o(1)$, which implies $\mathsf{TV}\,(\mathcal{P}, \mathcal{Q}) = o(1)$. Therefore, if $s^2 \leq \underline{C} \left( \frac{n \log n}{\log(1/(1-\rho^2))} \vee n \right)$, then $\mathsf{TV}\,(\mathcal{P}, \mathcal{Q}) \leq 0.1$. Moreover, if $s^2 \leq \underline{C} \left( \frac{n \log n}{\log(1/(1-\rho^2))} \vee n \right)$ or $s^2 = o(n)$, we have $\mathsf{TV}\,(\mathcal{P}, \mathcal{Q}) = o(1)$. This concludes the proof of Theorem 1.4.

## B. Proof of Propositions

### B.1. Proof of Proposition 2.2

We first upper bound $\mathcal{Q}\,(\mathcal{T} \geq \tau)$ under the null hypothesis $\mathcal{H}_0$ by the Chernoff bound and union bound. For any $X, Y \overset{\text{i.i.d.}}{\sim} \mathcal{N}(0,1)$ and $\lambda \in (0,1)$, we have

$$\begin{aligned}
\mathbb{E}\,[\exp\,(\lambda XY)] &= \int \int \frac{1}{2\pi} \exp\,(\lambda xy) \exp\left(-\frac{x^2 + y^2}{2}\right) dxdy \\
&= \int \int \frac{1}{2\pi} \exp\left(-\frac{1}{2}(x - \lambda y)^2\right) \exp\left(-\frac{1}{2}(1 - \lambda^2)y^2\right) dxdy \\
&= \int \int \frac{1}{2\pi} \exp\left(-\frac{z^2}{2}\right) \exp\left(-\frac{1}{2}(1 - \lambda^2)y^2\right) dzdy = \frac{1}{\sqrt{1 - \lambda^2}}.
\end{aligned} \tag{13}$$

Let $\lambda = \frac{\rho}{2}$. Recall that $\mathcal{S}_{s,m}$ denotes the set of injective mappings $\pi : S \subseteq V(G_1) \mapsto V(G_2)$ with $|S| = m$. For any $\pi \in \mathcal{S}_{s,m}$, $\mathsf{e}\left(\mathcal{H}_\pi^f\right) \sim \sum_{i=1}^{\binom{m}{2}} A_i B_i$, where $(A_i, B_i)$ are independent and identically distributed pairs of standard normals with correlation coefficient $\rho$. Then, by the Chernoff bound,

$$\begin{aligned}
\mathcal{Q}\left[\mathsf{e}\left(\mathcal{H}_\pi^f\right) \geq \tau\right] &\leq \exp\,(-\lambda\tau)\,\mathbb{E}\left[\exp\left(\lambda\mathsf{e}\left(\mathcal{H}_\pi^f\right)\right)\right] \\
&= \exp\,(-\lambda\tau)\,\mathbb{E}\left[\prod_{i=1}^m \exp\,(\lambda A_i B_i)\right] \\
&\overset{(a)}{\leq} \exp\left(-\lambda\binom{m}{2}\frac{\rho}{2} - \frac{1}{2}\binom{m}{2}\log\left(1 - \lambda^2\right)\right) \\
&= \exp\left(-\binom{m}{2}\left(\frac{\rho^2}{4} + \frac{1}{2}\log\left(1 - \frac{\rho^2}{4}\right)\right)\right) \overset{(b)}{\leq} \exp\left(-\frac{1}{12}\binom{m}{2}\rho^2\right),
\end{aligned} \tag{14}$$
$$\tag{15}$$

where (a) is because $\mathbb{E}\,[\lambda A_i B_i] = \frac{1}{\sqrt{1-\lambda^2}}$ for any $1 \leq i \leq \binom{m}{2}$; (b) follows from $\log(1-x) \geq -\frac{1}{3}x$ for $x = \frac{\rho^2}{4} \in \left[0, \frac{1}{4}\right]$. Applying the union bound, we obtain that

$$\mathcal{Q}\,(\mathcal{T} \geq \tau) \leq |\mathcal{S}_{s,m}|\,\mathcal{Q}\left[\mathsf{e}\left(\mathcal{H}_\pi^f\right) \geq \tau\right] \overset{(a)}{\leq} \binom{s}{m}^2 m! \exp\left(-\frac{1}{12}\binom{m}{2}\rho^2\right) \overset{(b)}{\leq} \exp\left(m\log\left(\frac{en}{1-\epsilon}\right) - \frac{1}{12}\binom{m}{2}\rho^2\right),$$

where (a) is because $|\mathcal{S}_{s,m}| = \binom{s}{m}^2 m!$ and (15); (b) is because $\binom{s}{m}m! \leq s^m$, $\binom{s}{m} \leq \left(\frac{e \cdot s}{m}\right)^m$ and $m = \frac{(1-\epsilon)s^2}{n}$. Consequently, when $m - 1 \geq \frac{24(1+\epsilon)\log\left(\frac{en}{1-\epsilon}\right)}{\rho^2}$, we have $\mathcal{Q}\left(\mathcal{T} \geq \tau\right) \leq \exp\left(-\epsilon m \log\left(\frac{en}{1+\epsilon}\right)\right) = o(1)$.

We then upper bound $\mathcal{P}\left(\mathcal{T} < \tau\right)$ under the alternative hypothesis $\mathcal{H}_1$. We note that

$$
\begin{aligned}
\mathcal{P}\left(\mathcal{T} < \tau\right) &\overset{(a)}{\leq} \mathcal{P}\left(|S_{\pi^*}| < m\right) + \mathcal{P}\left(\mathcal{T} < \tau, |S_{\pi^*}| \geq m\right) \\
&\overset{(b)}{\leq} \mathcal{P}\left(|S_{\pi^*}| < m\right) + \mathcal{P}\left(\mathcal{T} < \tau \,\middle|\, |S_{\pi^*}| \geq m\right) \\
&\overset{(c)}{\leq} \mathcal{P}\left(|S_{\pi^*}| < m\right) + \mathcal{P}\left(\mathsf{e}\left(\mathcal{H}_{\pi_m^*}^f\right) < \tau \,\middle|\, |S_{\pi^*}| \geq m\right) &(16) \\
&\overset{(d)}{\leq} \exp\left(-\frac{\epsilon^2 s^2}{2n}\right) + \exp\left(-\binom{m}{2}\frac{\rho^2}{4c_0^2}\right) + \exp\left(-\binom{m}{2}\frac{\rho}{2c_0}\right), &(17)
\end{aligned}
$$

where (a) is because of (5); (b) follows from $\mathcal{P}\left(\mathcal{T} < \tau, |S_{\pi^*}| \geq m\right) \leq \frac{\mathcal{P}(\mathcal{T} < \tau, |S_{\pi^*}| \geq m)}{\mathcal{P}(|S_{\pi^*}| \geq m)} = \mathcal{P}\left(\mathcal{T} < \tau \,\middle|\, |S_{\pi^*}| \geq m\right)$; (c) is because under the event $|S_{\pi^*}| \geq m$, there exists $\pi_m^* \in \mathcal{S}_{s,m}$ such that $\pi_m^* = \pi^*$ on its domain set $\mathrm{dom}(\pi_m^*)$; (d) uses the concentration (37) for Hypergeometric distribution and the Hanson-Wright inequality in Lemma D.1 with $M_0 = I_{\binom{m}{2}}$ and $\delta = \exp\left(-\binom{m}{2}\left(\frac{\rho^2}{4c_0^2} \wedge \frac{\rho}{2c_0}\right)\right)$, where $c_0$ is the universal constant in Lemma D.1. Consequently, we obtain that $\mathcal{P}\left(\mathcal{T} < \tau\right) = o(1)$ when $m - 1 \geq \frac{24(1+\epsilon)\log\left(\frac{en}{1-\epsilon}\right)}{\rho^2}$. Let $C_1 = 25$. Then, we have $\mathcal{P}\left(\mathcal{T} < \tau\right) + \mathcal{Q}\left(\mathcal{T} \geq \tau\right) = o(1)$ when $\frac{s^2}{n} - 1 \geq \frac{25\log n}{\rho^2}$ as $n$ becomes sufficiently large.

## B.2. Proof of Proposition 2.3

We first upper bound $\mathcal{Q}\left(\mathcal{T} \geq \tau\right)$ under the null hypothesis $\mathcal{H}_0$. We note that for any $X, Y \overset{\text{i.i.d.}}{\sim} \mathcal{N}(0, 1)$ and $\lambda > 0$, we have

$$
\begin{aligned}
\mathbb{E}\left[\exp\left(-\frac{\lambda}{2}(X - Y)^2\right)\right] &= \int\int \frac{1}{2\pi}\exp\left(-\frac{\lambda}{2}(x - y)^2\right)\exp\left(-\frac{1}{2}\left(x^2 + y^2\right)\right) dxdy \\
&= \int\int \frac{1}{2\pi}\exp\left(-\frac{\lambda+1}{2}\left(x - \frac{\lambda}{\lambda+1}y\right)^2\right)\exp\left(-\frac{2\lambda+1}{2(\lambda+1)}y^2\right) dxdy \\
&= \int\int \frac{1}{2\pi}\exp\left(-\frac{\lambda+1}{2}z^2\right)\exp\left(-\frac{2\lambda+1}{2(\lambda+1)}y^2\right) dydz = \frac{1}{\sqrt{1+2\lambda}}. &(18)
\end{aligned}
$$

Let $\lambda = \frac{1}{4(1-\rho)} - \frac{1}{2}$. Then, we have $1 + 2\lambda = \frac{1}{2(1-\rho)}$. Since $1 - e^{-6} < \rho < 1$, we also have $\lambda > 0$. Recall that $\mathcal{S}_{s,m}$ denotes the set of injective mappings $\pi : S \subseteq V(G_1) \mapsto V(G_2)$ with $|S| = m$. For any $\pi \in \mathcal{S}_{s,m}$, $\mathsf{e}\left(\mathcal{H}_\pi^f\right) \sim \sum_{i=1}^{\binom{m}{2}} -\frac{1}{2}(A_i - B_i)^2$, where $(A_i, B_i)$ are independent and identically distributed pairs of standard normals with correlation coefficient $\rho$. Then, by the Chernoff bound,

$$
\begin{aligned}
\mathcal{Q}\left(\mathsf{e}\left(\mathcal{H}_\pi^f\right) \geq \tau\right) &\leq \exp\left(-\lambda\tau\right)\mathbb{E}\left[\exp\left(\lambda\mathsf{e}\left(\mathcal{H}_\pi^f\right)\right)\right] \\
&\overset{(a)}{=} \exp\left(\binom{m}{2}\left(2(1-\rho)\lambda - \frac{1}{2}\log(1+2\lambda)\right)\right) \\
&= \exp\left(\binom{m}{2}\left(\frac{1}{2} - (1-\rho) - \frac{1}{2}\log\left(\frac{1}{2(1-\rho)}\right)\right)\right) \\
&\overset{(b)}{\leq} \exp\left(-\frac{\log(1/(1-\rho))}{3}\binom{m}{2}\right), &(19)
\end{aligned}
$$

where (a) follows from (18); (b) is because $\rho > 1 - e^{-6}$ implies that $\frac{1}{2} - (1-\rho) - \frac{1}{2}\log\left(\frac{1}{2(1-\rho)}\right) \leq 1 - \frac{1}{6}\log\left(\frac{1}{1-\rho}\right) - \frac{1}{3}\log\left(\frac{1}{1-\rho}\right) \leq -\frac{1}{3}\log\left(\frac{1}{1-\rho}\right)$. Then, applying the union bound yields that

$$
\mathcal{Q}\left(\mathcal{T} \geq \tau\right) \leq |\mathcal{S}_{s,m}|\mathcal{Q}\left[\mathsf{e}\left(\mathcal{H}_\pi^f\right) \geq \tau\right] \leq \exp\left(m\log\left(\frac{en}{1-\epsilon}\right) - \frac{1}{3}\log\left(\frac{1}{1-\rho}\right)\binom{m}{2}\right),
$$

where the last inequality is because $|\mathcal{S}_{s,m}| = \binom{s}{m}^2 m! \leq \left(\frac{es}{m}\right)^m s^m = \left(\frac{en}{1-\epsilon}\right)^m$. Therefore, when $m-1 \geq \frac{6(1+\epsilon)\log\left(\frac{en}{1+\epsilon}\right)}{\log(1/(1-\rho))}$, we have $\mathcal{Q}(\mathcal{T} \geq \tau) \leq \exp\left(-\epsilon m \log\left(\frac{en}{1-\epsilon}\right)\right)$.

We then upper bound $\mathcal{P}(\mathcal{T} < \tau)$ under the alternative hypothesis $\mathcal{H}_1$. By (16), we have that

$$\mathcal{P}(\mathcal{T} < \tau) \leq \mathcal{P}(|S_{\pi^*}| < m) + \mathcal{P}\left(\mathsf{e}\left(\mathcal{H}^f_{\pi_m^*}\right) < \tau \,\middle|\, |S_{\pi^*}| > m\right)$$

$$\leq \exp\left(-\frac{\epsilon^2 s^2}{2n}\right) + \exp\left(-\frac{1}{2}\binom{m}{2}(1-\log 2)\right),$$

where the last inequality follows from (37) in Lemma D.3 and $\frac{-\mathsf{e}\left(\mathcal{H}^f_{\pi_m^*}\right)}{1-\rho} \sim \chi^2\left(\binom{m}{2}\right)$, and the concentration inequality for chi-square distribution (34) in Lemma D.2. Since $m = \frac{(1-\epsilon)s^2}{n}$, there exists a universal constant $c_2 > 0$ such that, when $\frac{s^2}{n} \geq c_2$, we have $\mathcal{P}(\mathcal{T} < \tau) \leq 0.05$. Specifically, $\mathcal{P}(\mathcal{T} < \tau) = o(1)$ when $s^2/n = \omega(1)$. Since $\mathcal{Q}(\mathcal{T} \geq \tau) = o(1)$ when $m-1 \geq \frac{6(1+\epsilon)\log\left(\frac{en}{1+\epsilon}\right)}{\log(1/(1-\rho))}$, there exists a universal constant $C_2$ such that, when $s^2 \geq C_2\left(\frac{n\log n}{\log(1/(1-\rho))} \vee n\right)$, $\mathcal{P}(\mathcal{T} < \tau) + \mathcal{Q}(\mathcal{T} \geq \tau) \leq 0.1$. Specifically, when $s^2/n = \omega(1)$, we have $\mathcal{P}(\mathcal{T} < \tau) = o(1)$, and thus $\mathcal{P}(\mathcal{T} < \tau) + \mathcal{Q}(\mathcal{T} \geq \tau) = o(1)$.

*Remark* B.1. The Gaussian assumption on the weighted edges for $\beta_e(G_1)$ and $\beta_e(G_2)$ in Propositions 2.2 and 2.3 can be extended to the sub-Gaussian assumption. The main ingredients of our proof in these two Propositions are the analysis of the tail bound for the Gaussian distribution. We compute the Moment Generating Function (MGF) and use a standard Chernoff bound to bound $\mathcal{Q}(\mathcal{T} \geq \tau)$. Indeed, if we relax the distribution assumption to a sub-Gaussian distribution, since the linear sum of sub-Gaussian random variables remains a sub-Gaussian random variable, the MGF in (18) can be approximated by $\mathbb{E}\left[\exp(\lambda XY)\right] \leq \frac{1}{\sqrt{1-c_1\lambda^2}}$ for some constant $c_1 \in \mathbb{R}$. Since the product of sub-Gaussian random variables is a sub-exponential random variable, the MGF in (18) can be approximated by $\mathbb{E}\left[\exp\left(-\frac{\lambda}{2}(X-Y)^2\right)\right] \leq \frac{1}{\sqrt{1+c_2\lambda}}$ for some constant $c_2 \in \mathbb{R}$. For $\mathcal{P}(\mathcal{T} < \tau)$, the tail bound holds for sub-Gaussian as well.

### B.3. Proof of Proposition 3.1

For any $S \subseteq V(G_1)$ and $T \subseteq V(G_2)$ with $|S| = |T|$, define

$$\mathcal{P}(G_1, G_2, S, T) = \tilde{\mathcal{P}}(G_1[S], G_2[T]) \prod_{e \notin \binom{S}{2}} \mathcal{Q}_0(\beta_e(G_1)) \prod_{e \notin \binom{T}{2}} \mathcal{Q}_0(\beta_e(G_2)), \tag{20}$$

$$\mathcal{Q}(G_1, G_2, S, T) = \mathcal{Q}(G_1[S], G_2[T]) \prod_{e \notin \binom{S}{2}} \mathcal{Q}_0(\beta_e(G_1)) \prod_{e \notin \binom{T}{2}} \mathcal{Q}_0(\beta_e(G_2)), \tag{21}$$

where $G[S]$ for any $S \subseteq V(G)$ denotes the induced subgraph with vertex set $S$ of $G$; $\tilde{\mathcal{P}}$ denotes the distribution of two random graphs follow fully correlated Gaussian Wigner model; $\mathcal{Q}_0$ denotes the standard normal distribution. Recall that $S_{\pi^*} = V(G_1) \cap (\pi^*)^{-1}(V(G_2))$ and $T_{\pi^*} = \pi^*(V(G_1)) \cap V(G_2)$. Indeed, $\mathcal{P}(G_1, G_2, S, T)$ denotes the distribution under $\mathcal{P}$ when given $S_{\pi^*} = S$ and $T_{\pi^*} = T$. Besides, $\mathcal{Q}(G_1, G_2 | S, T)$ and $\mathcal{Q}(G_1, G_2)$ are the same distribution for any $S \subseteq V(G_1), T \subseteq V(G_2)$ with $|S| = |T|$.

Since $\mathcal{P}(\cdot | \mathcal{E}) = \frac{\mathcal{P}(\cdot, \mathcal{E})}{\mathcal{P}(\mathcal{E})} = \frac{\sum_{i=0}^{(1+\epsilon)s^2/n} \mathcal{P}(|S_{\pi^*}| = i)\mathcal{P}(\cdot \mid |S_{\pi^*}| = i)}{\mathcal{P}(\mathcal{E})}$ and $\mathsf{TV}\left(\sum_i \lambda_i \mathcal{P}_i, \mathcal{Q}\right) \leq \sum_i \lambda_i \mathsf{TV}(\mathcal{P}_i, \mathcal{Q})$ when $\sum_i \lambda_i = 1$, we obtain

$$\mathsf{TV}\left(\mathcal{P}'(G_1, G_2), \mathcal{Q}(G_1, G_2)\right) \leq \sum_{i=0}^{\frac{(1+\epsilon)s^2}{n}} \frac{\mathcal{P}(|S_{\pi^*}| = i)}{\mathcal{P}(\mathcal{E})} \cdot \mathsf{TV}\left(\mathcal{P}\left(G_1, G_2 \,\middle|\, |S_{\pi^*}| = i\right), \mathcal{Q}(G_1, G_2)\right). \tag{22}$$

For any $0 \leq i \leq \frac{(1+\epsilon)s^2}{n}$ and $S \subseteq V(G_1), T \subseteq V(G_2)$ with $|S| = |T| = i$, by the data processing inequality (see, e.g., Polyanskiy & Wu (2025, Section 3.5)), we have

$$\mathsf{TV}\left(\mathcal{P}\left(G_1, G_2 \,\middle|\, |S_{\pi^*}| = i\right), \mathcal{Q}(G_1, G_2)\right) \leq \mathsf{TV}\left(\mathcal{P}(G_1, G_2, S, T), \mathcal{Q}(G_1, G_2, S, T)\right)$$

$$= \mathsf{TV}\left(\tilde{\mathcal{P}}(G_1[S], G_2[T]), \mathcal{Q}(G_1[S], G_2[T])\right), \tag{23}$$

where the last equality follows from (20), (21) and the fact that $\mathsf{TV}(X \otimes Z, Y \otimes Z) = \mathsf{TV}(X, Y)$ for any distributions $X, Y, Z$ such that $Z$ is independent with $X$ and $Y$.

For the random graphs $G_1[S]$ and $G_2[T]$ with $S \subseteq V(G_1), T \subseteq V(G_2)$, and $|S| = |T|$, they follow the correlated Gaussian Wigner model with node set size $|S|$ under $\tilde{\mathcal{P}}$, while they are independent under $\mathcal{Q}$. It follows from Wu et al. (2023, Theorem 1) that, when $\frac{|S|}{\log|S|} \leq \frac{2}{\rho^2}$, the total variation distance $\mathsf{TV}\left(\tilde{\mathcal{P}}\left(G_1[S], G_2[T]\right), \mathcal{Q}\left(G_1[S], G_2[T]\right)\right) = o(1)$. We then verify the condition $\frac{|S|}{\log|S|} \leq \frac{2}{\rho^2}$ for any $0 \leq |S| \leq \frac{(1+\epsilon)s^2}{n}$. In fact, since $s^2 \leq \frac{n\log n}{2\log(1/(1-\rho^2))}$, we have

$$|S| \leq \frac{(1+\epsilon)s^2}{n} \leq \frac{(1+\epsilon)\log n}{2\log\left(1/(1-\rho^2)\right)} \leq \frac{2\log\left(1/\rho^2\right)}{\rho^2},$$

where the last inequality follows from $\log\left(1/(1-\rho^2)\right) \geq \rho^2$, $\frac{\log n}{2} < \log\left(1/\rho^2\right)$ and $\epsilon < 1$. Therefore, we obtain $\frac{|S|}{\log|S|} \leq \frac{\frac{2}{\rho^2}\log\left(1/\rho^2\right)}{\log(1/\rho^2)+\log(2\log(1/\rho^2))} \leq \frac{2}{\rho^2}$, and thus

$$\mathsf{TV}\left(\tilde{\mathcal{P}}\left(G_1[S], G_2[T]\right), \mathcal{Q}\left(G_1[S], G_2[T]\right)\right) = o(1)$$

for any $S \subseteq V(G_1), T \subseteq V(G_2)$ with $|S| = |T| \leq \frac{(1+\epsilon)s^2}{n}$. Combining this with (22) and (23), we conclude that

$$\mathsf{TV}\left(\mathcal{P}'\left(G_1, G_2\right), \mathcal{Q}\left(G_1, G_2\right)\right) \leq \sum_{i=0}^{\frac{(1+\epsilon)s^2}{n}} \frac{\mathcal{P}\left(|S_{\pi^*}| = i\right)}{\mathcal{P}(\mathcal{E})} \cdot \mathsf{TV}\left(\mathcal{P}\left(G_1, G_2 \big| |S_{\pi^*}| = i\right), \mathcal{Q}\left(G_1, G_2\right)\right)$$

$$\leq \sum_{i=0}^{\frac{(1+\epsilon)s^2}{n}} \frac{\mathcal{P}\left(|S_{\pi^*}| = i\right)}{\mathcal{P}(\mathcal{E})} \cdot o(1) = o(1).$$

Therefore,

$$\mathsf{TV}\left(\mathcal{P}(G_1, G_2), \mathcal{Q}(G_1, G_2)\right) \overset{(a)}{\leq} \mathsf{TV}\left(\mathcal{P}(G_1, G_2), \mathcal{P}'(G_1, G_2)\right) + \mathsf{TV}\left(\mathcal{P}'(G_1, G_2), \mathcal{Q}(G_1, G_2)\right)$$

$$\overset{(b)}{\leq} \mathsf{TV}\left(\mathcal{P}(G_1, G_2, \pi), \mathcal{P}'(G_1, G_2, \pi)\right) + \mathsf{TV}\left(\mathcal{P}'(G_1, G_2), \mathcal{Q}(G_1, G_2)\right)$$

$$= \mathcal{P}\left((G_1, G_2, \pi) \notin \mathcal{E}\right) + \mathsf{TV}\left(\mathcal{P}'(G_1, G_2), \mathcal{Q}(G_1, G_2)\right) = o(1), \tag{24}$$

where (a) follows from the triangle inequality and (b) is derived by the data processing inequality (see, e.g., Polyanskiy & Wu (2025, Section 3.5)).

### B.4. Proof of Proposition 3.5

Recall that the conditional distribution is defined as

$$\mathcal{P}'(G_1, G_2, \pi) = \frac{\mathcal{P}(G_1, G_2, \pi)\mathbb{1}_{(G_1,G_2,\pi)\in\mathcal{E}}}{\mathcal{P}(\mathcal{E})} = (1+o(1))\mathcal{P}\left(G_1, G_2, \pi\right)\mathbb{1}_{(G_1,G_2,\pi)\in\mathcal{E}},$$

where the last inequality holds because $\mathcal{P}(\mathcal{E}) = 1 - o(1)$. By (6) and (24), we have the following sufficient condition for the impossibility results:

$$\mathbb{E}_{\mathcal{Q}}\left[\left(\frac{\mathcal{P}'(G_1, G_2)}{\mathcal{Q}(G_1, G_2)}\right)^2\right] = 1 + o(1) \Rightarrow \mathsf{TV}\left(\mathcal{P}', \mathcal{Q}\right) = o(1) \Rightarrow \mathsf{TV}(\mathcal{P}, \mathcal{Q}) = o(1). \tag{25}$$

Recall the likelihood ratio in (7). To compute the conditional second moment, we introduce an independent copy $\tilde{\pi}$ of the latent permutation $\pi$ and express the square likelihood ratio as

$$\left(\frac{\mathcal{P}'(G_1, G_2)}{\mathcal{Q}(G_1, G_2)}\right)^2 = (1+o(1))\mathbb{E}_\pi\left[\frac{\mathcal{P}(G_1, G_2|\pi)}{\mathcal{Q}(G_1, G_2)}\mathbb{1}_{(G_1,G_2,\pi)\in\mathcal{E}}\right]\mathbb{E}_{\tilde{\pi}}\left[\frac{\mathcal{P}(G_1, G_2|\tilde{\pi})}{\mathcal{Q}(G_1, G_2)}\mathbb{1}_{(G_1,G_2,\tilde{\pi})\in\mathcal{E}}\right]$$

$$= (1+o(1))\mathbb{E}_{\pi\perp\tilde{\pi}}\left[\frac{\mathcal{P}(G_1, G_2|\pi)}{\mathcal{Q}(G_1, G_2)}\frac{\mathcal{P}(G_1, G_2|\tilde{\pi})}{\mathcal{Q}(G_1, G_2)}\mathbb{1}_{(G_1,G_2,\pi)\in\mathcal{E}}\mathbb{1}_{(G_1,G_2,\tilde{\pi})\in\mathcal{E}}\right].$$

Taking expectation for both sides under $\mathcal{Q}$, the conditional second moment is given by

$$
\begin{aligned}
\mathbb{E}_{\mathcal{Q}}\left[\left(\frac{\mathcal{P}'(G_1,G_2)}{\mathcal{Q}(G_1,G_2)}\right)^2\right] &= (1+o(1))\mathbb{E}_{\mathcal{Q}}\left[\mathbb{E}_{\pi\perp\tilde{\pi}}\left[\frac{\mathcal{P}(G_1,G_2|\pi)}{\mathcal{Q}(G_1,G_2)}\frac{\mathcal{P}(G_1,G_2|\tilde{\pi})}{\mathcal{Q}(G_1,G_2)}\mathbb{1}_{(G_1,G_2,\pi)\in\mathcal{E}}\mathbb{1}_{(G_1,G_2,\tilde{\pi})\in\mathcal{E}}\right]\right] \\
&= (1+o(1))\mathbb{E}_{\pi\perp\tilde{\pi}}\left[\mathbb{E}_{\mathcal{Q}}\left[\frac{\mathcal{P}(G_1,G_2|\pi)}{\mathcal{Q}(G_1,G_2)}\frac{\mathcal{P}(G_1,G_2|\tilde{\pi})}{\mathcal{Q}(G_1,G_2)}\mathbb{1}_{(G_1,G_2,\pi)\in\mathcal{E}}\mathbb{1}_{(G_1,G_2,\tilde{\pi})\in\mathcal{E}}\right]\right] \\
&= (1+o(1))\mathbb{E}_{\pi\perp\tilde{\pi}}\left[\mathbb{1}_{(G_1,G_2,\pi)\in\mathcal{E}}\mathbb{1}_{(G_1,G_2,\tilde{\pi})\in\mathcal{E}}\mathbb{E}_{\mathcal{Q}}\left[\frac{\mathcal{P}(G_1,G_2|\pi)}{\mathcal{Q}(G_1,G_2)}\frac{\mathcal{P}(G_1,G_2|\tilde{\pi})}{\mathcal{Q}(G_1,G_2)}\right]\right], \quad (26)
\end{aligned}
$$

where the last equality holds since $\mathcal{E}$ is independent with the edges in $G_1$ and $G_2$. Recall that $I^* = I^*(\pi,\tilde{\pi})$ defined in (10). Since $I^* = \cup_{C\in\mathsf{C}}\cup_{e\in C}\cup_{v\in V(e)\cap V(G_1)}v$ by the definition of $I^*$, we obtain that $\binom{I^*}{2} = \sum_{C\in\mathsf{C}}|C|$ by counting the edges induced by the vertices in $I^*$. Combining this with (9) and (26), we have that

$$
\begin{aligned}
\mathbb{E}_{\mathcal{Q}}\left[\left(\frac{\mathcal{P}'(G_1,G_2)}{\mathcal{Q}(G_1,G_2)}\right)^2\right] &= (1+o(1))\mathbb{E}_{\pi\perp\tilde{\pi}}\left[\mathbb{1}_{(G_1,G_2,\pi)\in\mathcal{E}}\mathbb{1}_{(G_1,G_2,\tilde{\pi})\in\mathcal{E}}\mathbb{E}_{\mathcal{Q}}\left[\frac{\mathcal{P}(G_1,G_2|\pi)}{\mathcal{Q}(G_1,G_2)}\frac{\mathcal{P}(G_1,G_2|\tilde{\pi})}{\mathcal{Q}(G_1,G_2)}\right]\right] \\
&= (1+o(1))\mathbb{E}_{\pi\perp\tilde{\pi}}\left[\mathbb{1}_{(G_1,G_2,\pi)\in\mathcal{E}}\mathbb{1}_{(G_1,G_2,\tilde{\pi})\in\mathcal{E}}\prod_{C\in\mathsf{C}}\left(\frac{1}{1-\rho^{2|C|}}\right)\right] \\
&\leq (1+o(1))\mathbb{E}_{\pi\perp\tilde{\pi}}\left[\mathbb{1}_{(G_1,G_2,\pi)\in\mathcal{E}}\mathbb{1}_{(G_1,G_2,\tilde{\pi})\in\mathcal{E}}\prod_{C\in\mathsf{C}}\left(\frac{1}{1-\rho^2}\right)^{|C|}\right] \\
&= (1+o(1))\mathbb{E}_{\pi\perp\tilde{\pi}}\left[\mathbb{1}_{(G_1,G_2,\pi)\in\mathcal{E}}\mathbb{1}_{(G_1,G_2,\tilde{\pi})\in\mathcal{E}}\left(\frac{1}{1-\rho^2}\right)^{|I^*|(|I^*|-1)/2}\right],
\end{aligned}
$$

where the inequality follows from $\frac{1}{1-\rho^{2x}} - \left(\frac{1}{1-\rho^2}\right)^x = \frac{(1-\rho^2)^x + (\rho^2)^x - 1}{(1-\rho^{2x})(1-\rho^2)^x} \leq \frac{1-\rho^2+\rho^2-1}{(1-\rho^{2x})(1-\rho^2)^x} = 0$ for any $0 < \rho < 1$ and $x \geq 1$. Since $\mathbb{P}\left[|I^*| = t\right] \leq \left(\frac{s}{n}\right)^{2t}$ by Lemma 3.4 and $|I^*| \leq |V(G_1)\cap\pi^{-1}(V(G_2))| \leq \frac{(1+\epsilon)s^2}{n}$ if $(G_1,G_2,\pi),(G_1,G_2,\tilde{\pi})\in\mathcal{E}$, we obtain

$$
\begin{aligned}
\mathbb{E}_{\mathcal{Q}}\left(\frac{\mathcal{P}'(G_1,G_2)}{\mathcal{Q}(G_1,G_2)}\right)^2 &\leq (1+o(1))\mathbb{E}_{\pi\perp\tilde{\pi}}\left[\mathbb{1}_{(G_1,G_2,\pi)\in\mathcal{E}}\mathbb{1}_{(G_1,G_2,\tilde{\pi})\in\mathcal{E}}\left(\frac{1}{1-\rho^2}\right)^{|I^*|(|I^*|-1)/2}\right] \\
&= (1+o(1))\sum_{t=0}^{\frac{(1+\epsilon)s^2}{n}}\mathbb{P}\left[|I^*| = t\right]\left(\frac{1}{1-\rho^2}\right)^{t(t-1)/2} \leq (1+o(1))\sum_{t=0}^{\frac{(1+\epsilon)s^2}{n}}\left(\frac{s}{n}\right)^{2t}\left(\frac{1}{1-\rho^2}\right)^{t(t-1)/2}.
\end{aligned}
$$

$$(27)$$

Let $a_t \triangleq \left(\frac{s}{n}\right)^{2t}\left(\frac{1}{1-\rho^2}\right)^{t(t-1)/2}$. For any $t < \frac{(1+\epsilon)s^2}{n}$, we have

$$
\frac{a_{t+1}}{a_t} = \frac{s^2}{n^2}\left(\frac{1}{1-\rho^2}\right)^t \leq \frac{s^2}{n^2}\left(\frac{1}{1-\rho^2}\right)^{\frac{(1+\epsilon)s^2}{n}} = \exp\left(\log\left(\frac{s^2}{n^2}\right) + \frac{(1+\epsilon)s^2}{n}\log\left(\frac{1}{1-\rho^2}\right)\right). \quad (28)
$$

Since $s^2 \leq \frac{n\log n}{8\log(1/(1-\rho^2))}$, we obtain

$$
\frac{(1+\epsilon)s^2}{n}\log\left(\frac{1}{1-\rho^2}\right) \leq \frac{(1+\epsilon)\log n}{8}
$$

and

$$
\log\left(\frac{s^2}{n^2}\right) \leq \log\left(\frac{\log n}{8n\log(1/(1-\rho^2))}\right) \overset{(a)}{\leq} -\frac{1}{2}\log n + \log\left(\frac{\log n}{8}\right),
$$

where (a) is because $\log\left(\frac{1}{1-\rho^2}\right) \geq \log\left(\frac{1}{1-n^{-1/2}}\right) \geq n^{-1/2}$. Combining this with (28), we obtain that $\frac{a_{t+1}}{a_t} \leq \exp\left(-\frac{(3-\epsilon)\log n}{8} + \log\left(\frac{\log n}{8}\right)\right) \leq n^{-1/4}$. Therefore, by (27),

$$\mathbb{E}_{\mathcal{Q}}\left(\frac{\mathcal{P}'(G_1, G_2)}{\mathcal{Q}(G_1, G_2)}\right)^2 \leq (1 + o(1)) \sum_{t=0}^{\frac{(1+\epsilon)s^2}{n}} a_t$$

$$= (1 + o(1)) \sum_{t=0}^{\frac{(1+\epsilon)s^2}{n}} \left(\frac{s}{n}\right)^{2t} \left(\frac{1}{1-\rho^2}\right)^{t(t-1)/2} \leq \frac{1 + o(1)}{1 - n^{-1/4}} = 1 + o(1),$$

which implies that $\mathsf{TV}(\mathcal{P}, \mathcal{Q}) = o(1)$ by (25).

## C. Proof of Lemmas

### C.1. Proof of Lemma 2.1

Recall that $|V(G_1)| = |V(G_2)| = s$ and $V(G_1) \subseteq V(\mathbf{G}_1), V(G_2) \subseteq V(\mathbf{G}_2)$. For any $\pi \sim \mathcal{S}_n$, we note that the event $|\pi(V(G_1)) \cap V(G_2)| = t$ with $t \in [s]$ can be divided as:

- Picking $t$ vertices in $V(G_1), V(G_2)$ respectively and constructing the mapping between picked vertices. We have $\binom{s}{t}^2 t!$ options for this step.

- Mapping the remaining $s - t$ vertices in $V(G_1)$ to $V(\mathbf{G}_2)\backslash V(G_2)$. We have $\binom{n-s}{s-t}(s-t)!$ options for this step.

- Mapping $V(\mathbf{G}_1)\backslash V(G_1)$ to the remaining vertices in $V(\mathbf{G}_2)$. We have $(n-s)!$ options for this step.

Then, for any $t \leq s$, we have that

$$\mathbb{P}\left[|\pi(V(G_1)) \cap V(G_2)| = t\right] = \frac{\binom{s}{t}^2 t! \cdot \binom{n-s}{s-t}(s-t)! \cdot (n-s)!}{n!} = \frac{\binom{s}{t}\binom{n-s}{s-t}}{\binom{n}{s}}, \tag{29}$$

which indicates that the size of intersection set $|\pi(V(G_1)) \cap V(G_2)|$ follows hypergeometric distribution $\mathsf{HG}(n, s, s)$ where $\pi \overset{\text{Unif.}}{\sim} \mathcal{S}_n$.

### C.2. Proof of Lemma 3.3

For any $P = (e_1, \pi(e_1), e_2, \cdots, e_j, \pi(e_j)) \in \mathsf{P}$ with $\tilde{\pi}(e_2) = \pi(e_1), \cdots, \tilde{\pi}(e_j) = \pi(e_{j-1})$, we have that

$$L_P = \prod_{i=1}^{j} \ell(e_i, \pi(e_i)) \prod_{i=2}^{j} \ell(e_i, \tilde{\pi}(e_i)) = \prod_{i=1}^{j} \ell(e_i, \pi(e_i)) \prod_{i=2}^{j} \ell(e_i, \pi(e_{i-1}))$$

$$= \ell(e_1, \pi(e_1))\ell(\pi(e_1), e_2) \cdots \ell(\pi(e_{j-1}), e_j)\ell(e_j, \pi(e_j)). \tag{30}$$

Under the distribution $\mathcal{Q}$, it follows from (30) that $L_P = \ell(B_0, B_1)\ell(B_1, B_2) \cdots \ell(B_{k-1}, B_k)$ for some $k \in \mathbb{N}$ and $B_0, B_1, \cdots, B_k \overset{\text{i.i.d.}}{\sim} \mathcal{N}(0, 1)$. Recall that

$$\ell(a, b) = \frac{\mathcal{P}\left(\beta_e(G_1) = a, \beta_{\pi(e)}(G_2) = b\right)}{\mathcal{Q}\left(\beta_e(G_1) = a, \beta_{\pi(e)}(G_2) = b\right)} = \frac{1}{\sqrt{1-\rho^2}} \exp\left(\frac{-\rho^2(a^2 + b^2) + 2\rho ab}{2(1-\rho^2)}\right), \text{ for any } a, b \in \mathbb{R}. \tag{31}$$

Then,

$$\mathbb{E}_{\mathcal{Q}}\left[L_P\right] = \mathbb{E}_{\mathcal{Q}}\left[\ell(B_0, B_1)\ell(B_1, B_2) \cdots \ell(B_{k-1}, B_k)\right]$$

$$= \frac{1}{(2\pi)^{(k+1)/2}\left((1-\rho^2)\right)^{k/2}} \int \cdots \int \exp\left(\sum_{t=0}^{k-1} \frac{-\rho^2(b_t^2 + b_{t+1}^2) + 2\rho b_t b_{t+1}}{2(1-\rho^2)}\right) \exp\left(\sum_{t=0}^{k} -\frac{b_t^2}{2}\right) db_0 \cdots db_k$$

$$= \frac{1}{(2\pi)^{(k+1)/2}\left((1-\rho^2)\right)^{k/2}} \int \cdots \int \exp\left(-\frac{\sum_{t=0}^{k-1}(b_t - \rho b_{t+1})^2}{2(1-\rho^2)} - \frac{b_k^2}{2}\right) db_0 \cdots db_k = 1,$$

where the last equality holds since the transformation $B'_t \triangleq \frac{B_t - \rho B_{t+1}}{\sqrt{1-\rho^2}}$ for any $0 \le t \le k-1$ yields that $\mathbb{E}_Q[L_P] = \frac{1}{(2\pi)^{(k+1)/2}} \int \cdots \int \exp\left(-\frac{\sum_{t=0}^{k-1} b_k'^2}{2} - \frac{b_k^2}{2}\right) db_0' \cdots db_{k-1}' db_k = 1$.

For any $C = (e_1, \pi(e_1), e_2, \cdots, e_j, \pi(e_j)) \in \mathsf{C}$ with $\tilde{\pi}(e_2) = \pi(e_1), \cdots, \tilde{\pi}(e_j) = \pi(e_{j-1})$ and $\tilde{\pi}(e_1) = \pi(e_j)$, we denote $e_0 = e_j$ for notational simplicity. Then, we have that

$$L_C = \prod_{i=1}^{j} \ell(e_i, \pi(e_i)) \prod_{i=1}^{j} \ell(e_i, \tilde{\pi}(e_i)) = \prod_{i=1}^{j} \ell(e_i, \pi(e_i)) \prod_{i=1}^{j} \ell(e_i, \pi(e_{i-1}))$$
$$= \ell(e_1, \pi(e_1))\ell(\pi(e_1), e_2) \cdots \ell(\pi(e_{j-1}), e_j)\ell(e_j, \pi(e_j))\ell(\pi(e_j), e_1).$$

Then $L_C = \ell(B_1, B_2) \cdots \ell(B_{k-1}, B_k)\ell(B_k, B_1)$ for $k = 2j$ and $B_1, \cdots, B_k \overset{\text{i.i.d.}}{\sim} \mathcal{N}(0,1)$. Denote $B_{k+1} = B_1$, we have that

$$\mathbb{E}_Q[L_C] = \mathbb{E}_Q\left[\ell(B_1, B_2) \cdots \ell(B_{k-1}, B_k)\ell(B_k, B_1)\right]$$
$$= \frac{1}{(2\pi(1-\rho^2))^{k/2}} \int \cdots \int \exp\left(\frac{\sum_{t=0}^{k-1} -\rho^2 (b_t^2 + b_{t+1}^2) + 2\rho b_t b_{t+1}}{2(1-\rho^2)}\right) \exp\left(\sum_{t=1}^{k} -\frac{b_t^2}{2}\right) db_1 \cdots db_k$$
$$= \frac{1}{(2\pi(1-\rho^2))^{k/2}} \int \cdots \int \exp\left(\frac{\sum_{t=0}^{k-1} -(b_t - \rho b_{t+1})^2}{2(1-\rho^2)}\right) db_1 \cdots db_k.$$

Let $C_t \triangleq B_t - \rho B_{t+1}$ for any $1 \le t \le k$. Then

$$[C_1, C_2, \cdots, C_{k-1}, C_k]^\top = \mathbf{J}_k [B_1, B_2, \cdots, B_{k-1}, B_k]^\top,$$

where

$$\mathbf{J}_k \triangleq \begin{bmatrix} 1 & -\rho & 0 & \cdots & 0 \\ 0 & 1 & -\rho & \cdots & 0 \\ 0 & 0 & 1 & \cdots & 0 \\ \vdots & \vdots & \vdots & \ddots & \vdots \\ -\rho & 0 & \cdots & 0 & 1 \end{bmatrix}$$

and thus $\det(\mathbf{J}_k) = 1 - \rho^k$ (see, e.g., Davis (1979, Section 3.2)). Then, we obtain that

$$\mathbb{E}_Q[L_C] = \frac{1}{(2\pi(1-\rho^2))^{k/2} \det(\mathbf{J}_k)} \int \cdots \int \exp\left(\frac{\sum_{t=1}^{k} -c_t^2}{2(1-\rho^2)}\right) dc_1 \cdots dc_k = \frac{1}{1-\rho^k} = \frac{1}{1-\rho^{2|C|}}.$$

## C.3. Proof of Lemma 3.4

Let $I' \triangleq \text{argmax}_{I \subseteq V(G_1), \pi(I) = \tilde{\pi}(I)} |I|$, we first show that $I' = I^*$. On the one hand, since $\pi(I') = \tilde{\pi}(I')$, we have $\pi\left(\binom{I'}{2}\right) = \tilde{\pi}\left(\binom{I'}{2}\right)$. Recall that the connected components of the correlated functional digraph in Definition 3.2 consist of paths and cycles. For any path $P \in \mathsf{P}$, we note that $\pi(P \cap V(G_1)) \ne \tilde{\pi}(P \cap V(G_1))$, and thus $\pi\left(\binom{P \cap V(G_1)}{2}\right) \ne \tilde{\pi}\left(\binom{P \cap V(G_1)}{2}\right)$. For any cycle $C \in \mathsf{C}$, we note that $\pi(C \cap V(G_1)) = \tilde{\pi}(C \cap V(G_1))$, and thus $\pi\left(\binom{C \cap V(G_1)}{2}\right) = \tilde{\pi}\left(\binom{C \cap V(G_1)}{2}\right)$. Therefore, $\binom{I'}{2} \subseteq \cup_{C \in \mathsf{C}} \cup_{e \in C \cap E(G_1)} e$. By the definition of $I^*$, we obtain $I' \subseteq I^*$. On the other hand, for any $C \in \mathsf{C}$, since $\pi\left(\cup_{e \in C \cap E(G_1)} e\right) = \tilde{\pi}\left(\cup_{e \in C \cap E(G_1)} e\right)$ by the definition of a cycle and $C \cap C' = \emptyset$ for any $C \ne C' \in \mathsf{C}$, we have that

$$\pi\left(\cup_{C \in \mathsf{C}} \cup_{e \in C \cap E(G_1)} e\right) = \tilde{\pi}\left(\cup_{C \in \mathsf{C}} \cup_{e \in C \cap E(G_1)} e\right).$$

Therefore, we have $\pi\left(\cup_{C \in \mathsf{C}} \cup_{e \in C} \cup_{v \in v(e) \cap V(G_1)} v\right) = \tilde{\pi}\left(\cup_{C \in \mathsf{C}} \cup_{e \in C} \cup_{v \in v(e) \cap V(G_1)} v\right)$, which implies $\pi(I^*) = \tilde{\pi}(I^*)$. Since $I^* \subseteq V(G_1)$, by the definition of $I'$, we conclude that $I^* \subseteq I'$. Therefore, we have $I^* = I' = \text{argmax}_{I \subseteq V(G_1), \pi(I) = \tilde{\pi}(I)} |I|$.

For any $t \leq s$, by the union bound, we obtain

$$\mathbb{P}\left[|I^*| = t\right] \leq \mathbb{P}\left[\exists A \subseteq V(G_1), |A| = t, \pi(A) = \tilde{\pi}(A) \subseteq V(G_2)\right]$$
$$\leq \binom{s}{t}\mathbb{P}\left[A \subseteq V(G_1), |A| = t, \pi(A) = \tilde{\pi}(A) \subseteq V(G_2)\right]. \tag{32}$$

For any fixed set $A \subseteq V(G_1)$ with $|A| = t$ and $\pi(A) = \tilde{\pi}(A) \subseteq V(G_2)$, we first choose a set $B \subseteq V(G_2)$ with $|B| = t$, and set $\pi(A) = \tilde{\pi}(A) = B$. There are $\binom{s}{t}$ ways to choose $B$, and $t!^2$ ways to map $\pi(A) = \tilde{\pi}(A) = B$. For the remaining vertices in $V(G_1)$, there are $(n-t)!^2$ ways to map them under $\pi$ and $\tilde{\pi}$. Therefore,

$$\binom{s}{t}\mathbb{P}\left[A \subseteq V(G_1), |A| = t, \pi(A) = \tilde{\pi}(A) \subseteq V(G_2)\right] = \binom{s}{t} \cdot \frac{1}{(n!)^2}\binom{s}{t}t!^2(n-t)!^2 \leq \left(\frac{s}{n}\right)^{2t}, \tag{33}$$

where the last inequality is due to the fact that $\binom{s}{t} \cdot \frac{1}{(n!)^2}\binom{s}{t}t!^2(n-t)!^2 = \left[\frac{s(s-1)\cdots(s-t+1)}{n(n-1)\cdots(n-t+1)}\right]^2$ and for any $i = 1, \cdots, t-1$, $\frac{s-i}{n-i} \leq \frac{s}{n}$. Combining this with (32), we obtain $\mathbb{P}\left[|I^*| = t\right] \leq \left(\frac{s}{n}\right)^{2t}$.

## D. Auxiliary Results

### D.1. Concentration Inequalities for Gaussian

**Lemma D.1** (Hanson-Wright inequality). *Let $X, Y \in \mathbb{R}^n$ be standard Gaussian vectors such that the pairs $(X_i, Y_i) \sim \mathcal{N}\left(\begin{pmatrix} 0 \\ 0 \end{pmatrix}, \begin{pmatrix} 1 & \rho \\ \rho & 1 \end{pmatrix}\right)$ are independent for $i = 1, \cdots, n$. Let $M_0 \in \mathbb{R}^{n \times n}$ be any deterministic matrix. There exists some universal constant $c_0 > 0$ such that*

$$\mathbb{P}\left[\left|X^\top M_0 Y - \rho \mathrm{Tr}(M_0)\right| \geq c_0 \left(\|M_0\|_F \sqrt{\log(1/\delta)} \vee \|M_0\|_2 \log(1/\delta)\right)\right] \leq \delta.$$

*Proof.* Note that $X^\top M_0 Y = \frac{1}{4}(X+Y)^\top M_0(X+Y) - \frac{1}{4}(X-Y)^\top M_0(X-Y)$ and

$$\mathbb{E}\left[(X+Y)^\top M_0(X+Y)\right] = (2+2\rho)\mathrm{Tr}(M_0), \mathbb{E}\left[(X-Y)^\top M_0(X-Y)\right] = (2-2\rho)\mathrm{Tr}(M_0).$$

By Hanson-Wright inequality (Hanson & Wright, 1971), there exists some universal constant $c_0$ such that

$$\mathbb{P}\left[\left|\frac{1}{4}(X+Y)^\top M_0(X+Y) - \frac{2+2\rho}{4}\mathrm{Tr}(M_0)\right| \geq \frac{c_0}{2}\left(\|M_0\|_F \sqrt{\log(1/\delta)} \vee \|M_0\|_2 \log(1/\delta)\right)\right] \leq \frac{\delta}{2},$$
$$\mathbb{P}\left[\left|\frac{1}{4}(X-Y)^\top M_0(X-Y) - \frac{2-2\rho}{4}\mathrm{Tr}(M_0)\right| \geq \frac{c_0}{2}\left(\|M_0\|_F \sqrt{\log(1/\delta)} \vee \|M_0\|_2 \log(1/\delta)\right)\right] \leq \frac{\delta}{2}$$

for any $\delta > 0$. Consequently,

$$\mathbb{P}\left[\left|X^\top M_0 Y - \rho \mathrm{Tr}(M_0)\right| \geq c_0 \left(\|M_0\|_F \sqrt{\log(1/\delta)} \vee \|M_0\|_2 \log(1/\delta)\right)\right]$$
$$\leq \mathbb{P}\left[\left|\frac{1}{4}(X+Y)^\top M_0(X+Y) - \frac{2+2\rho}{4}\mathrm{Tr}(M_0)\right| \geq \frac{c_0}{2}\left(\|M_0\|_F \sqrt{\log(1/\delta)} \vee \|M_0\|_2 \log(1/\delta)\right)\right]$$
$$+ \mathbb{P}\left[\left|\frac{1}{4}(X-Y)^\top M_0(X-Y) - \frac{2-2\rho}{4}\mathrm{Tr}(M_0)\right| \geq \frac{c_0}{2}\left(\|M_0\|_F \sqrt{\log(1/\delta)} \vee \|M_0\|_2 \log(1/\delta)\right)\right] \leq \delta.$$

$\square$

### D.2. Concentration Inequalities for Chi-Squared Distribution

**Lemma D.2** (Chernoff's inequality for Chi-squared distribution). *Suppose $\xi \sim \chi^2(n)$. Then, for any $\delta > 0$, we have*

$$\mathbb{P}\left[\xi > (1+\delta)n\right] \leq \exp\left(-\frac{n}{2}\left(\delta - \log\left(1+\delta\right)\right)\right), \tag{34}$$
$$\mathbb{P}\left[\xi < (1-\delta)n\right] \leq \exp\left(-\frac{n}{2}\left(-\delta - \log(1-\delta)\right)\right). \tag{35}$$

*Proof.* The results follow from Theorems 1 and 2 in Ghosh (2021). $\square$

### D.3. Concentration Inequalities for Hypergeometric Distribution

**Lemma D.3** (Concentration inequalities for Hypergeometric distribution). *For $\eta \sim \mathrm{HG}(n, s, s)$ and any $\epsilon > 0$, we have*

$$\mathbb{P}\left[\eta \geq \frac{(1+\epsilon)s^2}{n}\right] \leq \exp\left(-\frac{\epsilon^2 s^2}{(2+\epsilon)n}\right) \wedge \exp\left(-\frac{\epsilon^2 s^3}{n^2}\right), \tag{36}$$

$$\mathbb{P}\left[\eta \leq \frac{(1-\epsilon)s^2}{n}\right] \leq \exp\left(-\frac{\epsilon^2 s^2}{2n}\right) \wedge \exp\left(-\frac{\epsilon^2 s^3}{n^2}\right). \tag{37}$$

*Proof.* Denote $\xi \sim \mathrm{Bin}\left(s, \frac{s}{n}\right)$, by Theorem 4 in Hoeffding (1994), for any continuous and convex function $f$, we have

$$\mathbb{E}\left[f(\eta)\right] \leq \mathbb{E}\left[f(\xi)\right].$$

We note the function $f(x) = \exp(\lambda x)$ is continuous and convex for any $\lambda \in \mathbb{R}$. Therefore, we have $\mathbb{E}\left[\exp(\lambda \eta)\right] \leq \mathbb{E}\left[\exp(\lambda \xi)\right]$ for any $\lambda \in \mathbb{R}$, and thus the Chernoff bound for $\xi$ remains valid for $\eta$. Combining this with Theorems 4.4 and 4.5 in Mitzenmacher & Upfal (2005), we have

$$\mathbb{P}\left[\eta \geq \frac{(1+\epsilon)s^2}{n}\right] \leq \exp\left(-\frac{\epsilon^2 s^2}{(2+\epsilon)n}\right), \quad \mathbb{P}\left[\eta \leq \frac{(1-\epsilon)s^2}{n}\right] \leq \exp\left(-\frac{\epsilon^2 s^2}{2n}\right).$$

By Hoeffding's inequallity (Hoeffding, 1994), we also have

$$\mathbb{P}\left[\eta \geq \frac{(1+\epsilon)s^2}{n}\right] \leq \exp\left(-\frac{\epsilon^2 s^3}{n^2}\right), \quad \mathbb{P}\left[\eta \leq \frac{(1-\epsilon)s^2}{n}\right] \leq \exp\left(-\frac{\epsilon^2 s^3}{n^2}\right).$$

Therefore, we finish the proof of Lemma D.3. $\square$

## E. Additional Experiments

We provide a simple illustration on how our algorithm can be applied on real dataset. We conduct an experiment on Freeman's EIES networks (Freeman & Freeman, 1979), a small dataset of 46 researchers, where edge weights represent communication strength at two time points. We apply our method to test for correlation between these two temporal networks. We examine how sample size affects privacy protection by analyzing the normalized similarity score, defined as the similarity score $\mathsf{e}(\mathcal{H}_\pi^f)$ divided by $\binom{s}{2}$. Indeed, a lower score suggests weaker correlation and greater support for the null hypothesis of independence.

We apply our algorithm to the EIES dataset at different sample sizes, $s = 10, 20, 40$ and compute the corresponding normalized similarity scores: -1.066, -0.905, and -0.651. The scores increase with sample size, indicating stronger detected correlation. The lower scores at small sample sizes reflect failed correlation detection, quantifying the reduction in re-identification risk.

