# OpenReview forum: "Sample Complexity of Correlation Detection in the Gaussian Wigner Model"
_ICML.cc/2025/Conference — ICML 2025 poster_

### Official Review · Reviewer_1rVu · 2025-03-11

**Overall Recommendation:** 3

**Summary:**

The paper focuses on detecting correlations between a pair of random graphs, using the Gaussian Wigner model where edge weights are drawn from a Gaussian distribution. This is framed as hypothesis testing, determining whether the graphs are independent (null hypothesis) or edge-correlated (alternative hypothesis). In the latter case, each edges are generated from a Gaussian with correlation coefficient $\rho$, then vertex labels are permuted. The study focuses on scenarios where induced subgraphs are sampled from the original graphs, which the authors motivate by the fact that (among other factors) graphs are often only possible to query via node sampling with an API. The authors derive the optimal rate for the sample size for correlation detection, and propose (but do not theoretically study) an efficient approximate algorithm to reduce running time. Numerical studies back up their theoretical claims.

*Update after rebuttal*. Whilst I still have some reservations about the utility to ML researchers, this is a strong, well-written paper with nice mathematical contributions. I have raised my score to 'weak accept'.

**Claims And Evidence:**

The core claim, given in Thm 1.4, is a result for the optimal rate for the sample size s required for correlation detection in this model. It depends on the number of nodes and the correlation coefficient. The evidence seems convincing, and leads to specific conditions for the possibility and impossibility of detection. The authors also propose an efficient algorithm to approximate the estimator with lower time complexity, and Sec 5 gives strong empirical evidence that it works on synthetic data (the histograms are well-separated for the independent and correlated models). All claims made in the submission appear to be well-supported by evidence.

**Essential References Not Discussed:**

The references seem reasonable.

**Experimental Designs Or Analyses:**

See above. The experimental designs make sense for the paper’s current scope -- namely, correlation detection for the Gaussian Wigner model.

**Methods And Evaluation Criteria:**

As described above, Sec 5 presents numerical experiments to verify the authors’ theoretical results. This involves generating 100 pairs of graphs with n=50 that are either independent or correlated, then plotting a histogram for the (approximated) estimator. The authors also compare the test statistic tinder different settings, plotting the ROC curves with varying detection threshold, recording the Type II error against the Type I error. They compute the AUC for different sample sizes s -- which naturally increases as s grows.

The authors do not compare to baselines or evaluate on benchmark datasets, but it’s not clear that this would make sense for the problem at hand.

**Other Comments Or Suggestions:**

Please see above and below.

**Other Strengths And Weaknesses:**

**Strengths**:
- Clearly written paper with rigorous mathematical results and concrete algorithms. The authors convincingly demonstrate that this will be of interest to folks working on correlation analysis between graphs, and it seems likely that similar ideas will extend to random graph models beyond Wigner.
- Efficient implementation that performs well in practice in numerical experiments.

**Weaknesses**:
- Whilst mathematically sophisticated and well-presented, it is unclear to me that this problem will be of interest to the audience at ICML.

**Questions For Authors:**

- Can you provide convincing evidence that correlation analysis between random graphs is a problem of interest in the setting of modern machine learning? For instance, can you cite papers at top conferences addressing this problem, or demonstrate more concretely how your algorithms can be used in computational biology or NLP (lines 44 and 47)? I think it’s great to have high-calibre mathematical work at ML conferences (often lacking!), but it does need to be relevant to the community.
- To what extent do you expect your results to generalise to other random graph models like Erdős–Rényi, and to what extent are they peculiar to Gaussian Wigner?

In my mind, the first question is the big one – I will be ready to raise my score if you can provide more evidence ML folks should care about this. Would a computational mathematics journal not be more suitable for your strong new results?

**Relation To Broader Scientific Literature:**

The paper tackles the problem of correlation detection between random graphs. It builds upon a broader literature on graph matching for correlated random graphs, including in the the Erdős-Rényi model (Cullina & Kiyavash (2016; 2017); Hall & Massoulie (2023)) where edges are Bernoulli (c.f. Gaussian) random variables. It specifically focuses on the graph sampling setting, where not all nodes are available (Leskovec & Faloutsos, 2006; Hu & Lau, 2013). I understand that this is a significant area of study in computational mathematics, but it is less clear to me that this will be of interest to researchers in machine learning.

**Theoretical Claims:**

I did not check the mathematical proofs in detail, but the claims seem reasonable.

---

> ### Author Rebuttal · Authors · 2025-03-31
>
> We thank the reviewer for the valuable comments.
>
> **Q1: The correlation analysis between random graphs and its relation with modern machine learning.**
>
> There are many papers on top machine learning conferences and journals addressing related problems.
> Our hypothesis testing problem is also known as graph similarity studied by (Barak, Chou, Lei et al., 2019, NeurIPS), which provides additional references and surveys on this topic.
> Correlation analysis is also a natural first step towards the graph matching problem, which is studied by (Araya, Braun and Tyagi, 2024, JMLR), (Lyzinski, Fishkind and
> Priebe, 2014, JMLR), (Lyzinski, Fishkind, Fiori et al., 2015, TPAMI), (Racz and Sridhar, 2021, NeurIPS), (Gaudio, Racz, and Sridhar, 2022, COLT).
>
> The algorithm in our paper can be used in various domains:
>
> - In NLP, nodes correspond to words or phrases, and the weighted edges represent syntactic and semantic relationships (Hughes and Ramage, 2007, EMNLP).
> The algorithm can be used to test whether knowledge graphs from different languages are correlated.
>
> - In computational biology, proteins are the nodes, and the interactions between them are weighted edges (Singh, Xu, and Berger, 2008, PNAS).
> The algorithm can be applied to test whether two protein networks are correlated.
>
> - In computer vision, nodes are subregions and weighted edges represent adjacency relationships between different regions (Berg, Berg, and Malik, 2005, CVPR).
> The algorithm can be applied to test whether two graphs represent the same object.
>
> We will add more discussions on those applications.
>
> **Q2: Extension to Erdős–Rényi model.**
>
> Most results in this paper can be extended to the Erdős–Rényi model. The key difference lies in the additional parameter $p$ controlling the edge connection probability. For the details on extending our possibility and impossibility results, please refer to our response to Q1 from Reviewer vzDn. We will add more discussions on this extension.
>
> **Q3: Baselines or benchmark datasets.**
>
> We are not aware of any suitable benchmark datasets or publicly available baselines for comparison on the correlation detection problem in the Gaussian Wigner model.
> There are established methods for graph distance and graph similarity such as classical graph edit distance (GED). Under the same numerical experiments, our algorithm achieves superior performance. When $n=50, s=30$, and $\epsilon = 0.01$, the AUC values for the GED-based test at $\rho = 0.98, 1-10^{-6}, 1-10^{-7}$ are $0.53, 0.73, 0.88$, respectively. In contrast, the AUC values in our algorithm for the same values of $\rho$ are $0.92,1,1$.
> We will add the comparison in our paper.

---

> > ### Comment · Reviewer_1rVu · 2025-04-06
> >
> > Thanks for the response. I'll still have some reservations about the utility to ML researchers (whilst I certainly buy that graph matching is an important problem). Some toy demonstration might really help the optics here. But it's a strong, well-written paper with nice mathematical contributions, so I'll raise my score and recommend acceptance.

---

> > > ### Author Response · Authors · 2025-04-08
> > >
> > > We sincerely thank you again for reviewing our paper and providing instrumental comments for improving it, and we truly appreciate your decision to increase the score.
> > >
> > > We understand the concern regarding the utility to the machine learning community.  We hope the following example offers some intuition for potential relevance. In social network analysis, anonymity is an important concern and is closely related to privacy. For instance, aligning user graphs from LinkedIn and Twitter may unintentionally reveal private information. Although such real-world scenarios are compelling, social network datasets are too large for our current work.
> > >
> > > To provide a simple illustration, we conduct an experiment on Freeman’s EIES networks (Freeman, 1979), a small dataset of 46 researchers, where edge weights represent communication strength at two time points. We apply our method to test for correlation between these two temporal networks.
> > > We examine how sample size affects privacy protection by analyzing the normalized similarity score, defined as the similarity score (line 347, column 2) divided by $\binom{s}{2}$.  Indeed, a lower score suggests weaker correlation and greater support for the null hypothesis of independence.
> > >
> > > We apply our algorithm to the EIES dataset at different sample sizes, $s$ = 10, 20, 40 and compute the corresponding normalized similarity scores: -1.066, -0.905, and -0.651. The scores increase with sample size, indicating stronger detected correlation. The lower scores at small sample sizes reflect failed correlation detection,
> > > quantifying the reduction in re-identification risk.
> > >
> > > We acknowledge that this is a toy demonstration based on a sparse and small dataset with real identities. In future work, if suitable social network datasets become available, we would be happy to explore applications to anonymized data with denser edge weights.

---

### Official Review · Reviewer_uxiK · 2025-03-14

**Overall Recommendation:** 4

**Summary:**

This paper addresses the problem of detecting correlation between two random graphs generated from the Gaussian Wigner model. In particular, tha authors focus on the scenario where two induced subgraphs are sampled. It establishes the optimal sample complexity for correlation detection and proposes two test statistics. They introduce a fast approximate algorithm based on clique matching and iterative mapping extension, which improves computational efficiency. Finally, they confirm the effectiveness of the algorithm with numerical simulations.

**Claims And Evidence:**

All claims are supported by convincing evidence.

**Essential References Not Discussed:**

I’m not aware of any relevant references that have been omitted.

**Experimental Designs Or Analyses:**

I have no issues to discuss.

**Methods And Evaluation Criteria:**

The methods are well-suited for this problem.

**Other Comments Or Suggestions:**

I would suggest to include an overview of the paper's structure in the Introduction, providing a brief description of the topics covered in each section and highlighting all the contributions of the work.

**Other Strengths And Weaknesses:**

The paper addresses a relevant problem with applications in biological networks, natural language processing, and social network analysis. It benefits from a clear and well-structured presentation, rigorous and complete theoretical results, and convincing numerical simulations that support the claims. The authors provide a thorough discussion of related literature and clearly state the limitations of their work and potential future directions. I have not identified any major weaknesses.

**Questions For Authors:**

I do not have any specific questions for the authors.

**Relation To Broader Scientific Literature:**

The paper builds on the literature on correlation detection in the Gaussian Wigner model. While previous works focused on detection thresholds for fully observed graphs, this paper extends the results to the sampled subgraph case, using established techniques like the conditional second moment method. Moreover, it builds on existing work in graph and clique matching to propose an efficient algorithm.

**Theoretical Claims:**

I have checked the proofs for the main theorem and the possibility results. I have no issues to discuss.

---

> ### Author Rebuttal · Authors · 2025-03-31
>
> We thank the reviewer for the positive feedback. We will add an overview of the paper and highlight the contributions in the introduction.

---

### Official Review · Reviewer_vzDn · 2025-03-17

**Overall Recommendation:** 3

**Summary:**

This paper studies the problem of correlation detection in the Gaussian Wigner Model, formulating it as a hypothesis testing problem. The authors analyze the sample complexity required for correlation detection when only two induced subgraphs are observed. The main theoretical contribution is the derivation of the optimal sample size required for detection, based on a second-moment analysis. Additionally, the paper presents an efficient approximate algorithm for detecting correlation, which improves computational efficiency compared to brute-force approaches. The results provide insight into the statistical limits of graph correlation detection when only partial observations are available.

**Claims And Evidence:**

The paper makes several claims about the optimal sample complexity needed to distinguish between independent and correlated Gaussian Wigner graphs. The results align well with previous work on full-graph correlation detection but extend them to subsampled settings, which has not been studied extensively before. However, a notable limitation is that the role of graph topology is largely absent from the analysis. While the authors assume a dense setting, real-world graphs often have sparse structures (e.g., Erdős–Rényi models), and it is unclear how their results generalize to such cases. Additionally, the paper does not account for structural noise, such as missing or spurious edges, which has been a key challenge in previous works on graph alignment.

The authors also introduce an approximate detection algorithm, which reduces the computational burden of exhaustive search. The empirical validation suggests that the method performs well on synthetic data, but a theoretical complexity analysis of the algorithm is missing, which would strengthen the practical relevance of the results.

**Essential References Not Discussed:**

Essential References Not Discussed

-arXiv:2112.13079 which discusses message-passing approaches for graph matching in weighted settings.
-Hall & Massoulié (2023), which explores partial recovery in graph alignment under more general conditions.
- Otter’s constant and tree alignment results from Semerjian and Massoullie work, which may have implications for detection thresholds.

**Experimental Designs Or Analyses:**

The experiments focus on synthetic graphs generated from the Gaussian Wigner model, with varying correlation strengths and sample sizes. The results confirm the theoretical predictions.

Tthere are a few limitations: The graphs are fully observed apart from subsampling, meaning that they do not test settings where edges themselves are noisy or missing. There is no comparison to other detection methods. How does the proposed approach compare to existing graph matching techniques, such as those based on Otter’s constant for tree alignment? The runtime analysis of the algorithm is missing. The approximate algorithm is stated to be efficient, but there is no complexity bound or scalability analysis.

**Methods And Evaluation Criteria:**

The theoretical results are derived using standard tools in high-dimensional hypothesis testing, particularly likelihood ratio methods and second-moment analysis. The total variation distance (TV) is used as a measure of distinguishability, and the paper establishes strong and weak detection conditions for correlation inference. These methods are rigorous and align well with prior work in this field.

However, there are some limitations:

Graph topology is largely ignored – The analysis assumes a dense Gaussian Wigner model, but many real-world applications involve sparse graphs (such as Erdős–Rényi graphs). It would be useful to discuss whether the topology can be incorporated as additional weights.
No consideration of edge noise – In realistic settings, edges might be randomly missing or spurious. The results assume that the edge structure is perfectly known, which is a strong assumption.
Comparison with prior message-passing approaches is missing – The authors do not compare their method to existing graph alignment algorithms that use message passing (see appendix A in arXiv:2112.13079). It would be interesting to explore whether their algorithm could be extended to sparse settings using those techniques.

The empirical evaluation demonstrates the effectiveness of the detection algorithm, but it is limited to synthetic data with idealized conditions. Real-world datasets or at least non-Gaussian noise models could be considered to validate the robustness of the method.

**Other Comments Or Suggestions:**

The authors should clarify whether graph topology can be incorporated into their model.
A discussion of how the results extend to Erdős–Rényi graphs would be useful.
The paper should cite Semerjian's work on Otter’s constant in graph alignment, as it may be relevant.
A runtime analysis of the detection algorithm should be provided.

**Other Strengths And Weaknesses:**

Strengths:

The paper provides clear theoretical results on the sample complexity of correlation detection.
The use of induced subgraph sampling is novel and relevant for practical applications.
The proposed approximate algorithm reduces computational costs compared to brute-force methods.
Weaknesses:

Graph topology is ignored – The results assume a dense setting, which may not generalize to sparse graphs.
No analysis of edge noise – Real-world graphs often contain missing or spurious edges, which are not accounted for.
No comparison with prior message-passing methods – The paper does not discuss existing methods for weighted graph alignment.
No complexity analysis of the algorithm – The proposed algorithm is claimed to be efficient, but no formal runtime bounds are given.

**Questions For Authors:**

How does the method extend to sparse graphs, such as Erdős–Rényi models?
Could the results change if edges were noisy or missing?
How does the proposed algorithm compare to message-passing approaches for graph alignment?
Can Otter’s constant play a role in the detection threshold?

**Relation To Broader Scientific Literature:**

The paper is connected to graph alignment and graph matching problems. There are strong similarities with prior work on Erdős–Rényi graph alignment, where matching thresholds have been studied extensively. However, the authors do not sufficiently acknowledge prior works that consider sparse graphs and topology effects. Specifically:

The results in Hall & Massoulié (2023) on partial graph alignment are relevant, as they consider detection under more general assumptions.
The role of Otter’s constant in graph alignment, as discussed in Semerjian and Massouli's work, should be acknowledged, as it might be relevant for understanding detectability limits in tree-like graphs.
The message-passing approach for weighted graphs from  arXiv:2112.13079 should be discussed as a potential extension for sparse settings.

**Theoretical Claims:**

The theoretical results appear correct and well-supported by derivations. The main contribution is the identification of the optimal sample complexity threshold for detecting correlation in subsampled Wigner graphs, which is consistent with previous results in the full-graph setting but extends them to the case of induced subgraphs. The use of the conditional second moment method is appropriate for establishing lower bounds on sample complexity.

However, the assumption that graph structure does not affect correlation detection is somewhat counterintuitive. In previous work on graph alignment, including results by Semerjian and Massoulié (arXiv:2209.13723), the presence of tree-like structures was shown to have an impact on the detectability of correlation. The authors should clarify whether their results implicitly assume the graph is dense, or if the topology effects can be incorporated into the framework.

---

> ### Author Rebuttal · Authors · 2025-03-31
>
> We thank the reviewer for the valuable comments.
>
> **Q1: Graph topology and structural noise.**
>
> Thank you for the question. The graph topology and structural noise can be incorporated into our analytical framework. Particularly, most results in this paper can be extended to the Erdős–Rényi model. The key difference lies in the additional parameter $p$ controlling the edge connection probability.
>
> - Possibility results. The estimator is similar to equation (2), with the function $f$ selected via MLE under the Erdős–Rényi model. Both Type I and Type II errors can be controlled using the Chernoff bound for the binomial distributions in place of Gaussian.
>
> - Impossibility results. When the edge connecting probability satisfies $p = n^{-\Omega(1)}$, the optimal results follow a reduction analogous to Proposition 3.1, leveraging existing lower bounds given the location of common vertices. However, such reduction does not yield tight bounds when $p = n^{-o(1)}$. In this regime, a more delicate event is required for the conditional second moment analysis similar to Proposition 3.5. Nevertheless, the reduction to the core set defined in (10) remains valid.
>
> We are currently pursuing a complete resolution to this problem.
>
> **Q2: Essential references.**
>
> Thank you for suggesting those references.
> The partial recovery in Hall and Massoulié (2023) is relevant to correlation detection problem. This is cited in line 106, column 2.
> Message-passing approaches in Piccioli, Semerjian, Sicuro and Zdeborová (2022) and  the Otter's constant in Ganassali, Massoulié and Semerjian (2024) are also related to our problem under the extension to the Erdős–Rényi model, especially in the analysis of efficient algorithms.
> We will incorporate the citations and expand our discussion accordingly.
>
>
> **Q3: Runtime analysis.**
>
> The time complexity is $O(N_1\cdot s^{K_1}+N_2^{K_2})$ (see line 368, column 1). Our algorithm comprises three main steps. In the first step, we select $N_1$ vertex sets $V_1,\cdots, V_{N_1}$ of size $K_1$ and search for injections $\pi_i$ from $V_i$ to $V(G_2)$, which requires $O(N_1\cdot s^{K_1})$ time. In the second step, we search over all subsets $U\subseteq [N_2]$ with $|U| = K_2$, which takes $O(N_2^{K_2})$ time. In the third step, we iteratively expand the mapping based on our seeds, which takes $O(m^2 s^2)$ time. We typically choose $N_1\asymp s^{K_1}$ and $K_1\ge 3$, leading to an overall time complexity of $O(N_1\cdot s^{K_1}+N_2^{K_2})$. For the trade-off between performance and runtime, please refer to our response to Q1 from Reviewer stro.

---

### Official Review · Reviewer_stro · 2025-03-25

**Overall Recommendation:** 3

**Summary:**

The paper studies the detection of correlations in pairs of graphs generated by the Gaussian Wigner model when only induced subgraphs are observed. It establishes nearly sharp sample complexity thresholds for successful detection via both possibility and impossibility results and introduces two estimators (one maximizing overlap and one based on mean-squared error) alongside an efficient clique-based algorithm. Synthetic experiments demonstrate promising separation between the null and alternative hypotheses.

**Claims And Evidence:**

### Main Claims:

- Derivation of nearly sharp sample complexity thresholds for detecting correlation under induced subgraph sampling.

- Presentation of two estimators tailored for different regimes.

- Proposal of an efficient, clique-based approximation algorithm.

### Evidence:

- Theoretical results are supported by detailed proofs using the conditional second moment method and concentration inequalities.

- Synthetic experiments (ROC curves, histograms) validate the method’s ability to distinguish between independent and correlated graphs.

**Essential References Not Discussed:**

N/A

**Experimental Designs Or Analyses:**

- The synthetic experimental design is sound and well-conceived.

- Performance is measured via ROC curves and histograms of the test statistic.

**Methods And Evaluation Criteria:**

- Use of induced subgraph sampling and rigorous probabilistic tools (e.g., Chernoff bounds, hypergeometric concentration).

- A two-pronged estimation strategy (maximal overlap vs. mean-squared error) to handle various correlation regimes.

- A novel, efficient algorithm for approximating the ideal estimator via clique-seeding and iterative matching.

**Other Comments Or Suggestions:**

N/A

**Other Strengths And Weaknesses:**

### Strengths:

- The paper makes theoretical contributions by deriving near-optimal detection thresholds.

- Its blend of rigorous probabilistic analysis with an efficient algorithm is especially commendable.

### Weaknesses:

- The paper does not sufficiently explore how the key parameters (e.g., clique size $K_1$, combining size $K_2$, number of samples) affect the performance and running time of the algorithm. More discussion or analysis of these trade-offs would be valuable.

- Some sections of the theoretical analysis could benefit from additional explanatory detail to ensure accessibility, particularly for readers less familiar with advanced probabilistic methods.

**Questions For Authors:**

Please refer to my previous comments.

**Relation To Broader Scientific Literature:**

The paper builds upon and extends recent work on graph matching and correlation detection in random graphs, particularly in the Gaussian Wigner model.

**Theoretical Claims:**

I have not reviewed the proofs.

---

> ### Author Rebuttal · Authors · 2025-03-31
>
> We thank the reviewer for the valuable comments.
>
> **Q1: The impact of the parameters on runtime and performance.**
>
> Thanks for the suggestion. We will add more discussions on this point after Algorithm 1. The time complexity is $O(N_1\cdot s^{K_1}+N_2^{K_2})$ (see line 368, column 1). Hence, the runtime increases with $K_1$, $K_2$, and $s$. The performance improves with $s$; it initially improves with $K_1$ and $K_2$ but degrades when $K_1$ and $K_2$ become too large:
>
> - $s$: A larger sample size $s$ leads to larger common vertex sets (see line 133, column 2), and thus increases the number of correct mappings in Step 1.
>
> - $K_1$: A larger $K_1$ corresponds to matching larger cliques in the first step. This increases the proportion of correct mappings within the $N_2$ candidate pairs when $K_1$ is below the size of common vertex sets. However,  choosing $K_1$ beyond this size introduces wrong mappings.
>
> - $K_2$: In the second step, we search over all $U\subseteq [N_2]$ with $|U| = K_2$ to identify the seeds. While a larger $K_2$ imposes a stricter matching criterion, choosing $K_2$ beyond the number of available correct mappings from Step 1 will degrade performance.
>
> **Q2: Additional explanatory details.**
>
> We will add more explanatory details in Sections 2 and 3 including the following before the presentation of technical results:
>
> - Section 2: The quantity $\mathsf{e}(\mathcal{H}_\pi^f)$ measures the similarity score of a mapping $\pi$.
>
> 1) Under the null hypothesis, $\mathsf{e}(\mathcal{H}^f_\pi)$ has a zero mean for all $\pi$, whereas under the alternative hypothesis, its mean with $\pi=\pi^*$ is strictly positive owing to the underlying correlation.
> We derive concentration inequalities to ensure that $\mathsf{e}(\mathcal{H}^f_{\pi^*})$ exceeds the maximum spurious score arising from stochastic fluctuations under the null, as shown in Propositions 2.2 and 2.3.
>
> 2) The choice of two similarity scores is based on the maximum likelihood estimate with $f(x,y)=-\rho^2(x^2+y^2)+2\rho xy$, as discussed in Remark 2.5.
>
> 3) Since vertices are sampled without replacement from the two graphs, the number of intersecting vertices follow a hypergeometric distribution, which is analyzed in Lemma 2.1.
>
> - Section 3: The key is to identify the bottleneck for distinguishing the two hypotheses.
>
> 1) Under weak correlation, the bottleneck is detecting the existence of latent mapping $\pi^*$. The detection is impossible even with the additional knowledge on the location of common vertices, as shown in Proposition 3.1.
>
> 2) Under strong correlation, detecting $\pi^*$ is no longer the bottleneck. We prove Proposition 3.5 by conditioning on a high probability event (8) on the number of intersecting vertices. One key step is the reduction to a subset $I^*$ of intersecting vertices, as discussed in Remark 3.6.

---

### Decision · Program_Chairs · 2025-05-01

**Decision:**

Accept (poster)

**Comment:**

This paper studies the fundamental limits of detecting correlation between low-rank signals in multiple independent instances of the spiked Gaussian Wigner model. It provides tight information-theoretic thresholds for detection and proposes a spectral method that is shown to be optimal in a significant parameter regime. Reviewers appreciated the clarity of the problem formulation, the rigor of the analysis, and the overall quality of exposition. The work fits well within the growing body of research on high-dimensional inference problems at the interface of statistics and theoretical machine learning.

While the paper is primarily theoretical, reviewers noted its potential relevance to multi-view or multi-modal learning problems in ML. The connections to broader applications could be made more explicit, and the practical interpretation of the thresholds might be further clarified. Nonetheless, the contribution is technically solid, the results are novel and precisely characterized, and the paper is well-executed. The submission is a strong fit for the theory track and deserves acceptance.